# Decoupling Reasoning and Confidence:
# Resurrecting Calibration in Reinforcement Learning from Verifiable Rewards

**Zhengzhao Ma** [1 2]  **Xueru Wen** [1 2]  **Boxi Cao** [1]  **Yaojie Lu** [1]  **Hongyu Lin** [1]  **Jinglin Yang** [3 4 5]  **Min He** [5]
**Xianpei Han** [1 2]  **Le Sun** [1 2]

## Abstract

Reinforcement Learning from Verifiable Rewards (RLVR) significantly enhances large language models (LLMs) reasoning but severely suffers from calibration degeneration, where models become excessively over-confident in incorrect answers. Previous studies devote to directly incorporating calibration objective into existing optimization target. However, our theoretical analysis demonstrates that there exists a fundamental gradient conflict between the optimization for maximizing policy accuracy and minimizing calibration error. Building on this insight, we propose DCPO, a simple yet effective framework that systematically decouples reasoning and calibration objectives. Extensive experiments across mathematical reasoning and code generation benchmarks demonstrate that our DCPO not only preserves accuracy on par with GRPO but also achieves the best calibration performance and substantially mitigates the over-confidence issue. Our study provides valuable insights and practical solution for more reliable LLM deployment.

## 1. Introduction

Reinforcement Learning from Verifiable Rewards (Lambert et al., 2024) have recently emerged as a cornerstone for shaping the remarkable reasoning abilities of large language models (Guo et al., 2025; Jaech et al., 2024). By

[1]Chinese Information Processing Laboratory, Institute of Software, Chinese Academy of Sciences, Beijing, China [2]University of Chinese Academy of Sciences, Beijing, China [3]Institute of Information Engineering, Chinese Academy of Sciences, Beijing, China [4]School of Cyber Security, University of Chinese Academy of Sciences, Beijing, China [5]National Computer Network Emergency Response Technical Team/Coordination Center of China, Beijing, China. Correspondence to: Boxi Cao <caoboxi@iscas.ac.cn>, Min He <hemin@cert.org.cn>.

*Proceedings of the 43$^{rd}$ International Conference on Machine Learning*, Seoul, South Korea. PMLR 306, 2026. Copyright 2026 by the author(s).

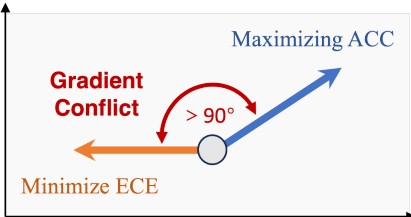

*Figure 1.* Illustration of gradient conflict between policy accuracy maximization and calibration error minimization.

optimizing policies using automatically verifiable rewards with on-policy sampling, representative RLVR algorithms such as GRPO (Shao et al., 2024) have achieved remarkable advances in mathematical reasoning (Hu et al., 2025), code generation (Luo et al., 2025a) and question answering (Chen et al., 2025; Wen et al., 2025) tasks.

Despite the success, RLVR often leads to severe *calibration degeneration*, emerging as a critical bottleneck that limits the practical applicability of LLMs in real-world scenarios (Bereket & Leskovec, 2025; Chhikara, 2025; Kirichenko et al., 2025). In particular, RL-trained models tend to become **over-confident**, assigning extremely high confidence to their output even when the answers are incorrect. In high-stakes domains such as healthcare, law, and finance, such over-confidence can mislead users about the system's reliability, resulting in amplified systemic risk. (Yao et al., 2025; Omar et al., 2024; Guan et al., 2024) To address this challenge, previous studies (Damani et al., 2025; Xu et al., 2024; Leng et al., 2024) attempt to jointly optimize correctness and calibration by incorporating calibration objectives into existing RL optimization targets. Unfortunately, such paradigm often leads to "accuracy-calibration tradeoff", where improvements in calibration come at the expense of reasoning accuracy. Therefore, how to reliably improve the calibration of LLMs without compromising their reasoning capability has become an urgent challenge to ensure their trustworthy deployment in real-world applications.

To investigate the underlying causes of the above issues in RLVR, we conduct a series of theoretical analyses[1] of the

---

[1]A concise theoretical analysis is provided in Section 4, with the complete mathematical derivation deferred to the appendix A.

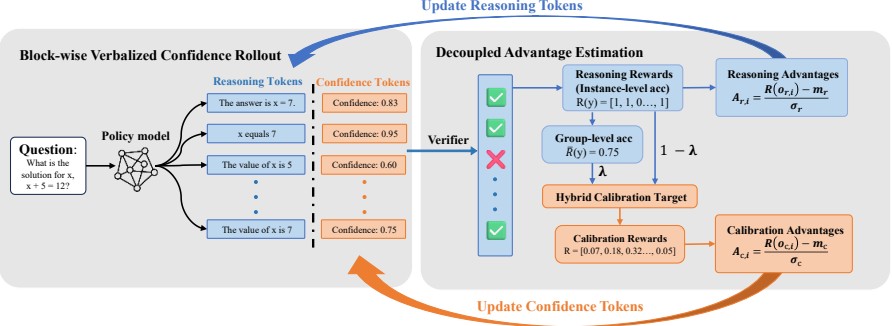

*Figure 2.* The overall framework of DCPO, which leverages block-wise verbalized confidence rollout and decoupled advantage estimation to decouple the optimization objectives of accuracy and calibration, and further integrates instance-level and group-level signals for more stable calibration optimization.

prevailing optimization paradigms, thereby uncovering the key factors that lead to the "accuracy–calibration tradeoff". Specifically, we reveal a critical gradient conflict between accuracy and calibration. As illustrated in Figure 1, for over-confident models trained with RLVR, the gradient direction for maximizing accuracy is *negatively aligned* with the gradient direction for minimizing calibration error. That is, the Fisher-metric inner product between these two gradients is negative, implying an optimization conflict. This finding sheds light on why previous approaches suffered from the accuracy–calibration tradeoff, since naively coupling these objectives struggles to reach a Pareto-optimal state. Moreover, we find that traditional instance-level binary supervision is insufficient for calibration optimization, as it is highly stochastic, and drives the model toward overly sharp probability distributions. The above analysis indicates that the key to jointly optimizing accuracy and calibration in RLVR lies in decoupling the optimization gradients and introducing more informative supervision signals.

Building on the above insights, we propose *Decoupled Calibration Policy Optimization*(DCPO), a simple yet effective framework that systematically decouples reasoning accuracy and calibration objectives at the levels of generation structure, reward design, and gradient optimization. Specifically, as demonstrated in Figure 2, DCPO requires the model to explicitly verbalize its confidence after reasoning trajectory generation. We assign separate rewards to the reasoning tokens and the confidence tokens, and apply a masked gradient strategy to ensure that the two parts of the sequence are optimized toward distinct objectives. In this way, DCPO fundamentally avoids the accuracy–calibration gradient conflict and enables parallel improvement of both reasoning ability and confidence reliability. Furthermore, to construct stable and low-variance supervision for calibration optimization, we exploit the group sampling mechanism inherent in GRPO (Shao et al., 2024). Specifically, we prove that, although the correctness of individual instances is not precisely distinguished, the average correctness within a rollout group provides a more stable estimate of the model's

uncertainty for a given input. Accordingly, we supervise confidence prediction using both instance-level and group-level accuracy, without requiring any additional annotation or external oracle. This joint signal yields consistent and low-variance calibration feedback during training.

To comprehensively evaluate the effectiveness of DCPO, we conduct experiments on both mathematical reasoning and code generation tasks. For mathematical reasoning, we evaluate on 5 widely-used benchmarks spanning a broad range of difficulty levels, including MATH (Hendrycks et al., 2021), AIME, and AMC (Yang et al., 2024a). For Code generation, we evaluate on 3 widely-used benchmarks, including LiveCodeBench (Jain et al., 2024), and HumanEval+ (Liu et al., 2023). The results demonstrate that, compared with previous baselines, DCPO consistently achieves the best tradeoff between reasoning performance and calibration across domains. Specifically, across five mathematical benchmarks, Qwen3-8B trained with DCPO achieves an average accuracy improvement of 11.8%, matching the performance of vanilla GRPO and outperforming RLCR by 4.3% and CCGSPG by 3.2%. Meanwhile,DCPO achieves a relative Expected Calibration Error (ECE) reduction of 71.6% compared with QWEN3-8B, decreasing it from 0.435 to 0.128.

Our main contributions are summarized as follows[2]

- We identify a fundamental mechanism underlying calibration degeneration in RLVR and formalize the gradient conflict between accuracy and calibration.

- We introduce DCPO, a simple yet effective framework that systematically decouples reasoning and calibration objectives.

- Experiments demonstrate that DCPO achieves the best tradeoff between reasoning performance and calibration compared with previous strong baselines.

---

[2]The code is available at https://github.com/icip-cas/DCPO.

## 2. Preliminaries and Related Work

This section first introduces a widely used RLVR algorithm, Group Relative Policy Optimization (Shao et al., 2024), and then reviews the commonly used methods for measuring LLM confidence and calibration.

### 2.1. Group Relative Policy Optimization (GRPO)

GRPO (Shao et al., 2024) is an outcome-based RLVR algorithm that normalize rewards within a sampled response group. Given a prompt $q$, the policy $\pi_\theta$ samples a group of $G$ responses $\{o_i\}_{i=1}^G$, each assigned a scalar reward $r_i$. GRPO computes a group-relative advantage

$$A_i = \frac{r_i - m}{\sigma}, \quad m = \frac{1}{G}\sum_{i=1}^G r_i, \quad \sigma^2 = \frac{1}{G}\sum_{i=1}^G (r_i - m)^2. \quad (1)$$

Group normalization reduces reward scale sensitivity and provides a low-variance signal, making GRPO widely adopted in recent RLVR pipelines.

### 2.2. Confidence Estimation

Model confidence measures a language mode's self-assessed probablity that its generated answer is correct (Yoon et al., 2025). Existing approaches can be categorized into 3 classes (Moskvoretskii et al., 2025; Heo et al., 2024). **Token-based confidence** derives confidence from internal generation statistics like token probabilities (Fomicheva et al., 2020; Kadavath et al., 2022; Li et al., 2025a). While effective, it often requires additional computation and lacks interpretability. In this work, we estimate token-based confidence with commonly used sequence probability (Zheng et al., 2025; Liu et al., 2025; Fomicheva et al., 2020; Grabinski et al., 2022): $\text{Conf}(y) = \prod_{i=1}^{|y|} \pi_\theta(y_i \mid q, y_{<i})$. **Verbalized confidence** prompts models to explicitly output a confidence score alongside the answer (Lin et al., 2022; Xiong et al., 2023; Yang et al., 2024b), offering a flexible and human-interpretable interface (Yoon et al., 2025). **Consistency-based confidence** estimates uncertainty via agreement among multiple sampled outputs (Lin et al., 2023; Ding et al., 2025), which is considered to excel in downstream performance at the cost of additional sampling (Moskvoretskii et al., 2025).

### 2.3. Calibration Estimation

Calibration measures the alignment between predicted confidence and empirical correctness (Guo et al., 2017). A commonly used metric is Expected Calibration Error (ECE) (De-Groot & Fienberg, 1983), defined as

$$ECE = \sum_{m=1}^M \frac{|B_m|}{N}\big|a(B_m) - c(B_m)\big|, \quad (2)$$

where $B_m$ denotes a confidence bin with average confidence $c(B_m)$ and accuracy $a(B_m)$. AUROC (Hanley & McNeil, 1982) is commomly used to measure the discriminative quality of confidence scores. During reinforcement learning, ECE may decrease trivially as accuracy improves. To better characterize over-confidence, we additionally report *Positive Calibration Error (PCE)*, which restricts ECE to bins where confidence exceeds accuracy. Formally, PCE is defined as:

$$\text{PCE}(c, r) = \sum_{c(B_m) > a(B_m)} \frac{|B_m|}{N}\big|a(B_m) - c(B_m)\big|. \quad (3)$$

### 2.4. Calibration Optimization Methods

Existing calibration optimization methods fall into 2 categories. **Post-hoc and inference-time methods** (Wang et al., 2025; Li et al., 2025b) optimize calibration independently of training, e.g., inference-time distractors (Chhikara, 2025) or external confidence predictors (Ni et al., 2025). **Calibration-aware RL methods** (Zhou et al., 2025; Xu et al., 2024) integrate uncertainty objectives into policy optimization. For instance, CCGSPG (Liu et al., 2025) modifies GRPO objective according to token-based confidence, and RLCR (Damani et al., 2025) incorparates a Brier Score loss into RLVR rewards. These methods improve calibration but often entangle confidence learning with correctness optimization, leading to accuracy–calibration tradeoffs.

## 3. Empirical Analysis for Calibration Degeneration

In this section, we find that current LLMs consistently suffer from over-confidence issue, and this over-confidence issue is exacerbated by RLVR. Furthermore, we demonstrate that existing coupled optimization approaches cannot resolve this issue without noticeably degrading model performance.

### 3.1. Over-Confidence of Current LLMs

To systematically quantify the calibration behavior of existing LLMs, we conduct a comprehensive evaluation across different model families and scales on a suite of mathematical reasoning benchmarks.

Figure 3 reports the reliability diagrams of these LLMs on the combined AMC23 and AMC24 datasets. The results demonstrate mis-calibration is pervasive across all evaluated base models and is primarily driven by systematic over-confidence. Specifically, all models exhibit large ECE, with values consistently exceeding 0.3, indicating severe deviation from ideal calibration. Moreover, across most confidence bins, the empirical accuracy bars lie substantially below the diagonal, showing that models frequently assign high confidence to incorrect answers.

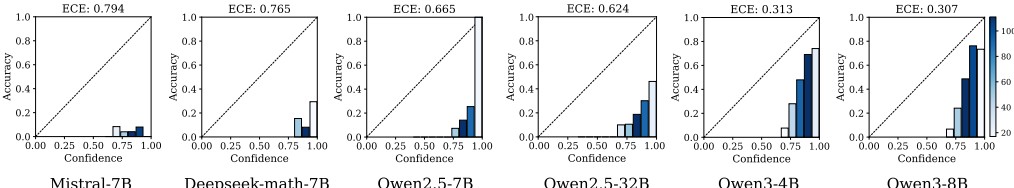

*Figure 3.* Reliability diagrams for different LLMs. The dashed line denotes perfect calibration; bar height indicates empirical accuracy per confidence bin, and color intensity reflects sample frequency. The Expected Calibration Error (ECE) is reported above each subplot, revealing prevalent over-confidence across models.

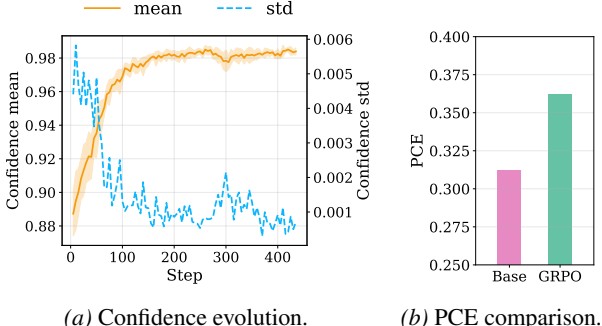

*(a)* Confidence evolution.   *(b)* PCE comparison.

*Figure 4.* The impact of RLVR on LLM calibration, which demonstrate that model confidence increases during RLVR training and RLVR exacerbates the models' over-confidence.

### 3.2. RLVR Leads to Calibration Degradation

To systematically examine how reinforcement learning with verifiable rewards (RLVR) affects model calibration, we conduct GRPO training on Qwen3-8B (non-thinking) (Yang et al., 2025). The model is trained on the DeepScaler dataset (Luo et al., 2025b), and calibration performance is evaluated on AIME24 using 8 repeated samplings to obtain stable estimates.

Figure 4 summarizes the evolution of model confidence and over-confidence during RLVR training. Figure 4(a) tracks the average predicted confidence over training steps and demonstrates a clear confidence upward trend. In particular, GRPO steadily increases the model's average confidence from about 0.88 to above 0.98, and decrease model confidence variance from 0.006 to about 0.001. Figure 4(b) further quantifies this phenomenon using PCE, which isolates a substantial amplification of over-confidence. In particular, after GRPO training, PCE increases from 0.312 to 0.362.

Together, these results show that, although RLVR improves reasoning accuracy, it simultaneously degrades calibration by pushing the model toward excessively confident predictions, even when answers are incorrect. This empirical evidence highlights a fundamental limitation of correctness-only RL optimization and motivates the need for calibration-aware training strategies that explicitly control confidence during reinforcement learning.

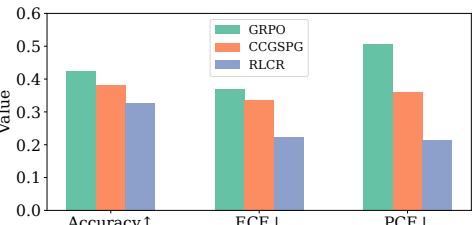

*Figure 5.* The accuracy and calibration performance of QWEN3-8B trained with different RL methods.The figures illustrate that while existing calibration optimization methods can improve model calibration, their accuracy decreases.

### 3.3. Accuracy-Calibration Tradeoff of Coupled Optimization

Recent calibration-aware reinforcement learning methods aim to jointly optimize reasoning accuracy and confidence calibration by directly incorporating calibration-related objectives into the RL reward or advantage function. To study the impact of coupled optimization objectives on model performance and calibration, we compare standard GRPO with two representative coupled calibration methods, RLCR (Damani et al., 2025) and CCGSPG (Liu et al., 2025), on the AIME24 benchmark. For each method, we evaluate reasoning accuracy together with calibration metrics, including ECE and PCE, enabling a systematic analysis of the accuracy–calibration relationship under coupled optimization.

Figure 5 summarizes the results, which indicates that such coupled optimization fundamentally impact reasoning performance. In particular, both RLCR and CCGSPG substantially reduce ECE and PCE compared to GRPO, indicating improved calibration, while both methods exhibit a noticeable reasoning accuracy performance drop relative to GRPO. This suggests that directly coupling calibration objectives with correctness-driven optimization introduces gradient interference, where enforcing conservative confidence estimates suppresses the learning signal for correct reasoning.

## 4. Theoretical Analysis

The experimental results in Section 3 imply that RLVR methods introduces severe over-confidence problem and existing

coupled optimization methods exhibit accuracy–calibration tradeoff. To dive into the underlying causes of the above issues, we firstly analyze why current RLVR exacerbates over-confidence. Furthermore, we reveal an accuracy-calibration gradient conflict, which explains why previous coupled optimization are plagued by accuracy-calibration tradeoff. Finally, we show that group-level accuracy can serve as a low-variance supervision signal for calibration training, inspiring the design of the calibration reward in DCPO. Collectively, these theoretical analyses provide a solid foundation for the design of our DCPO algorithm. We present concise proofs in this section and detailed proofs in Appendix A.

### 4.1. Trajectory-Level Reinforcement Learning Induces Over-Confidence

*Findings 1. The over-confidence phenomenon is a structural consequence of trajectory-level RLVR.*

For a fixed input $x$, let $\pi_\theta(y \mid x)$ denote the probability of generating a complete response trajectory $y$. Let $R(y) \in \{0, 1\}$ be a trajectory-level correctness indicator and define

$$\mathcal{Y}^+ := \{y \mid R(y) = 1\} \tag{4}$$

The expected accuracy objective optimized by trajectory-level reinforcement learning is

$$J_{\mathrm{acc}}(\theta) := \mathbb{E}_{y \sim \pi_\theta(\cdot \mid x)}[R(y)] = \sum_{y \in \mathcal{Y}^+} \pi_\theta(y \mid x) \tag{5}$$

**Proposition 4.1** (Mode Collapse). *In the absence of explicit entropy regularization, any optimal solution to $\max_\theta J_{\mathrm{acc}}(\theta)$ assigns probability mass $1$ to a single trajectory $y^* \in \mathcal{Y}^+$.*

*Proof.* Let $\Delta(\mathcal{Y})$ denote the probability simplex over the finite or countable set of all possible trajectories $\mathcal{Y}$. Then find the optimizal solution to objective $J_{\mathrm{acc}}(\theta)$ is equivalent to solve

$$\max_{p \in \Delta(\mathcal{Y})} \sum_{y \in \mathcal{Y}^+} p(y). \tag{6}$$

Obviously, objective 6 is a linear programming problem on $\Delta(\mathcal{Y})$, which is a convex polygonal set of $p$. By the fundamental theorem of linear programming, the optimal solution to this question is approached on a extreme point $\delta_{y^*}$, where $\delta_{y^*}(y) = \mathbb{I}[y = y^*]$. As $R(y) \in \{0, 1\}$, we must have $R(y^*) = 1$ at the optimal solution.

A detailed proof is provided in Appendix A.1 □

We further note that these low-entropy solutions are stable under small perturbations of the input. Since the policy logits are continuous functions of $x$, extreme logit margins at training inputs induce neighborhoods in input space where

$$\max_y \pi_\theta(y \mid x') \approx 1 \tag{7}$$

As correctness is discontinuous with respect to the input, this leads to over-confident yet incorrect predictions under distribution shift.

### 4.2. Accuracy-Calibration Gradient Conflict

*Findings 2. The gradient direction for maximizing accuracy is negatively aligned with the gradient direction for minimizing calibration error.*

Define the model confidence prediction as

$$\mathrm{Conf}_\theta(x) := \sum_{y \in \mathcal{Y}} \pi_\theta(y \mid x)\, \phi(y), \tag{8}$$

where $\phi : \mathcal{Y} \to [0, 1]$ is a trajectory confidence feature. We assume the confidence and model accuracy are positively related. Formally, this is expressed as:

$$\mathrm{Cov}_{\pi_\theta}(R(y), \phi(y)) > 0. \tag{9}$$

Consider a calibration objective of the form

$$J_{\mathrm{cal}}(\theta) := -\ell(\mathrm{Conf}_\theta(x),\, \mathbb{E}_{y \sim \pi_\theta}[R(y)]), \tag{10}$$

where $\ell$ is a proper scoring rule (Gneiting & Raftery, 2007).

**Proposition 4.2** (Gradient Conflict). *If the model is over-confident:* $\mathrm{Conf}_\theta(x) > \mathbb{E}_{y \sim \pi_\theta}[R(y)]$, *then the ascent direction of $J_{\mathrm{acc}}$ and $J_{\mathrm{cal}}$ satisfy*

$$\left\langle \nabla_\theta J_{\mathrm{acc}}(\theta),\, \nabla_\theta J_{\mathrm{cal}}(\theta) \right\rangle_{F^{-1}} < 0. \tag{11}$$

*Proof.* Firstly, we compute these two gradients as:

$$\nabla_\theta J_{\mathrm{acc}}(\theta) = \mathbb{E}_{y \sim \pi_\theta}[(R - \mathbb{E}[R])g] \tag{12}$$

$$\nabla_\theta J_{\mathrm{cal}}(\theta) = -\frac{\partial l}{\partial c} \mathbb{E}_{y \sim \pi_\theta}[(\phi - \mathbb{E}[\phi])g] \tag{13}$$

Then we derive the Fisher-metric inner product of these two gradients as:

$$\left\langle \nabla_\theta J_{\mathrm{acc}}(\theta),\, \nabla_\theta J_{\mathrm{cal}}(\theta) \right\rangle_{F^{-1}} = -\frac{\partial l}{\partial c}\, \mathrm{Cov}_{\pi_\theta}(R(y), \phi(y)). \tag{14}$$

Which is strictly negative due to assumption 9.

A detailed proof is provided in Appendix A.2 □

### 4.3. Group-Level Accuracy as Low-Variance Supervision

*Findings 3. The average correctness within a rollout group provides a more stable estimate of the model's uncertainty for a given input.*

Consider the group-level accuracy estimator

$$\tilde{R}_G := \frac{1}{G} \sum_{i=1}^{G} R(y_i). \tag{15}$$

**Proposition 4.3.** $\tilde{R}_G$ *is an unbiased estimator of* $\mathbb{E}[R(y)]$ *with variance* $O(1/G)$.

*Proof.* By the definition of $\tilde{R}_G$, the unbiaseness follows:

$$\mathbb{E}[\tilde{R}_G] = \mathbb{E}[R(y)] \qquad (16)$$

and the variance decay follows:

$$Var(\tilde{R}_G) = \frac{1}{G} Var(R(y)). \qquad (17)$$

A detailed proof is provided in Appendix A.4 □

**Proposition 4.4.** *When using the absolute calibration loss* $\ell(c, r) = |c - r|$, *using* $\tilde{R}_G$ *as supervision reduces gradient variance compared to instance-level correctness.*

*Proof.* We consider $R(y)$ as i.i.d Bernoulli random variable with expectation $p = \tilde{R}_G$. Let $g(c, r)$ denote an arbitrary measurable selection from $\partial_c |c - r|$. For any such seleection, we have $|g(c, r)| \leq 1$. When $c \in (0, 1)$ and $c \neq \tilde{R}_G$, we have:

$$Var(g(c, R(y))) = 4p(p - 1). \qquad (18)$$

$$Var(g(c, \tilde{R}_G)) = 0. \qquad (19)$$

Thus, $Var(g(c, R(y))) \geq Var(g(c, \tilde{R}_G))$, and the equality is approached iff $p \in \{0, 1\}$.

A detailed proof is provided in Appendix A.4 □

# 5. Decoupled Calibration Policy Optimization

To address the accuracy-calibration tradeoff induced by RLVR, based on the above analysis, we propose **DCPO** (**D**ecoupled **C**alibration **P**olicy **O**ptimization), a decoupled reward modeling framework for calibrated LLM training.

As shown in Figure 2, our approach separates the optimization of reasoning quality and confidence estimation by assigning distinct rewards and advantages to different segments of the model output. To provide a low-variance supervision signal for confidence calibration, we leverage the group sampling mechanism in GRPO, enabling stable and effective training without sacrificing task performance.

## 5.1. Block-wise Verbalized Confidence Rollout

To explicitly model uncertainty during reinforcement learning, we adopt a block-wise verbalized confidence rollout that separates the model output into a reasoning block and a confidence block. Specifically, given an input $q$, we prompt the model to generate a response $o$ in the following structured form $o = [\, o_r \;\; \texttt{<conf>} \;\; o_c \,]$, where $o_r$ contains the reasoning process and final answer, and $o_c$ is a scalar confidence prediction, with the two segments separated by a special delimiter token $\texttt{<conf>}$.

## 5.2. Decoupled Advantage Estimation

To optimize model calibration independently of accuracy, we design decoupled advantage estimators for reasoning and calibration.

For the reasoning component, we employ a standard outcome-based reward defined by accuracy:

$$R(o_r) = \mathbb{I}(y_{\text{pred}} \equiv y_{\text{label}}), \qquad (20)$$

where $y_{\text{pred}}$ is extracted from $o_r$.

Based on the analysis in 4.3, we exploit the group sampling mechanism inherent in GRPO and introduce a group-level accuracy $\tilde{R}_G = \frac{1}{G} \sum_{i=1}^{G} R(o_{r,i})$ as a calibration target.

To balance stability and expressiveness, we introduce a hybrid calibration target that interpolates between group-level and instance-level accuracy:

$$R_{IG} = \lambda \cdot \tilde{R}_G + (1 - \lambda) \cdot R(o_r), \qquad (21)$$

where $\lambda \in [0, 1]$ controls the tradeoff between variance reduction and sample-level discrimination. Accordingly, we define the confidence reward as

$$R_c(o_c) = -\left| \text{confidence}(o_c) - R_{IG} \right|. \qquad (22)$$

To ensure structural compliance, we apply a penalty when the model fails to follow the prescribed output format.

The resulting accuracy and confidence advantage are defined as:

$$A_{r,i} = \frac{R(o_{r,i}) - m_r}{\sigma_r}, A_{c,i} = \frac{R_c(o_{c,i}) - m_c}{\sigma_c}, \qquad (23)$$

where $m_r, m_c$ and $\sigma_r, \sigma_c$ denote the mean and standard deviation of corresponding rewards within the group.

## 5.3. Masked Gradient Optimization

To prevent interference between correctness optimization and confidence calibration, we adopt a masked gradient optimization strategy that applies distinct advantage signals to different token blocks.

For each response $o_i$, we construct a token-level mask that separates reasoning tokens $o_r$ from confidence tokens $o_c$. During policy optimization, advantages signals are applied exclusively to their corresponding token subsets:

$$\frac{1}{G} \sum_{i=1}^{G} \frac{1}{|o_i|} \left[ \sum_{y_j \in o_r} \hat{\rho}_{i,j} A_{r,i} + \sum_{y_j \in o_c} \hat{\rho}_{i,j} A_{c,i} \right], \qquad (24)$$

where $\hat{\rho}_{i,j}$ denotes the clipped importance sampling ratio for token $y_j$.

This block-wise masked optimization ensures that gradients derived from correctness supervision do not affect confidence estimation, and vice versa, enabling effective decoupling of reasoning accuracy and calibration objectives under a shared policy.

*Table 1.* Accuracy and calibration evaluation on 6 mathematical reasoning benchmarks. We report Accuracy (Acc) for reasoning performance, ECE, PCE and AUROC for calibration estimation. DCPO-I and DCPO-G denote variants that use only instance-level accuracy and only group-level accuracy, respectively, as calibration optimization signals. Bold and underlined values indicate the best and second-best results in each column, respectively.

| Method | Confidence | MATH-500 | | | | AIME24 | | | | AIME25 | | | |
|---|---|---|---|---|---|---|---|---|---|---|---|---|---|
| | | Acc↑ | ECE↓ | PCE↓ | AUROC↑ | Acc↑ | ECE↓ | PCE↓ | AUROC↑ | Acc↑ | ECE↓ | PCE↓ | AUROC↑ |
| Base | logtis | 84.7 | 0.075 | 0.078 | 0.743 | 27.5 | 0.461 | 0.431 | 0.750 | 20.0 | 0.486 | 0.483 | 0.796 |
| Base | verbal | 82.8 | 0.115 | 0.114 | 0.566 | 25.8 | 0.534 | 0.550 | 0.674 | 18.3 | 0.665 | 0.624 | 0.612 |
| GRPO | logtis | 90.4 | 0.054 | 0.104 | 0.726 | **42.5** | 0.370 | 0.505 | 0.736 | 29.2 | 0.469 | 0.556 | 0.742 |
| GRPO | verbal | 88.0 | 0.105 | 0.104 | 0.475 | 40.0 | 0.515 | 0.510 | 0.556 | 28.3 | 0.593 | 0.593 | 0.568 |
| ConfClass | - | 90.4 | 0.102 | 0.079 | 0.524 | 42.5 | 0.363 | 0.297 | 0.642 | 29.2 | 0.401 | 0.352 | 0.523 |
| RLCR | verbal | 90.2 | 0.051 | 0.047 | 0.782 | 32.8 | 0.224 | 0.214 | 0.823 | 24.1 | 0.134 | 0.138 | 0.621 |
| CCGSPG | logtis | 88.6 | 0.057 | 0.082 | 0.853 | 38.3 | 0.337 | 0.360 | 0.833 | 27.5 | 0.457 | 0.459 | 0.794 |
| DCPO | verbal | 90.2 | 0.049 | 0.033 | 0.861 | 41.6 | 0.188 | 0.212 | **0.914** | 28.3 | **0.130** | **0.133** | **0.951** |
| DCPO-G | verbal | **90.8** | **0.038** | **0.029** | **0.874** | 41.6 | 0.330 | 0.307 | 0.852 | **30.0** | 0.433 | 0.418 | 0.836 |
| DCPO-I | verbal | 89.7 | 0.068 | 0.037 | 0.864 | 40.0 | **0.170** | **0.178** | 0.831 | 28.3 | 0.141 | 0.146 | 0.723 |

| Method | Confidence | AMC23 | | | | AMC24 | | | | **Overall Performance** | | | |
|---|---|---|---|---|---|---|---|---|---|---|---|---|---|
| | | Acc↑ | ECE↓ | PCE↓ | AUROC↑ | Acc↑ | ECE↓ | PCE↓ | AUROC↑ | Acc↑ | ECE↓ | PCE↓ | AUROC↑ |
| Base | logtis | 63.2 | 0.204 | 0.255 | 0.832 | 49.4 | 0.329 | 0.315 | 0.759 | 49.0 | 0.311 | 0.312 | 0.796 |
| Base | verbal | 61.8 | 0.370 | 0.344 | 0.567 | 43.3 | 0.492 | 0.498 | 0.627 | 46.4 | 0.435 | 0.426 | 0.609 |
| GRPO | logtis | 76.1 | 0.160 | 0.277 | **0.850** | 63.9 | 0.264 | 0.370 | 0.693 | 60.4 | 0.248 | 0.362 | 0.749 |
| GRPO | verbal | 72.0 | 0.277 | 0.259 | 0.519 | 58.9 | 0.370 | 0.350 | 0.544 | 57.4 | 0.372 | 0.363 | 0.532 |
| ConfClass | - | 76.1 | 0.196 | 0.154 | 0.623 | 63.9 | 0.293 | 0.226 | 0.565 | 60.4 | 0.271 | 0.222 | 0.575 |
| RLCR | verbal | 75.0 | 0.143 | 0.105 | 0.736 | 60.5 | **0.132** | 0.142 | 0.807 | 56.5 | 0.139 | 0.128 | 0.753 |
| CCGSPG | logtis | 71.1 | 0.118 | 0.210 | 0.825 | 62.3 | 0.187 | 0.305 | 0.772 | 57.6 | 0.230 | 0.283 | 0.815 |
| DCPO | verbal | 75.6 | 0.092 | 0.089 | 0.849 | **65.0** | 0.191 | 0.194 | 0.832 | **60.8** | **0.128** | 0.126 | **0.881** |
| DCPO-G | verbal | **76.4** | **0.073** | **0.075** | 0.847 | 63.4 | 0.186 | 0.237 | 0.822 | 60.5 | 0.209 | 0.229 | 0.846 |
| DCPO-I | verbal | 71.9 | 0.134 | 0.116 | 0.828 | 64.7 | 0.140 | **0.133** | 0.847 | 58.7 | 0.138 | **0.122** | 0.819 |

**Theorem 5.1** (Statistical Optimality of Decoupled Calibration). *Under a strictly proper scoring rule, the optimal confidence predictor satisfies*

$$\mathbb{E}[c \mid q] = \mathbb{E}_{y \sim \pi_\theta(\cdot|q)}[R(y)]. \tag{25}$$

*Proof.* Consider $R(y)$ as i.i.d Bernoulli distribution. For a proper calibration estimation function $\ell(c, r)$, the expected loss should be minimized at $c = r$, and the expected loss

$$\arg\min_c \mathbb{E}_{r \sim Bern(p)}[l(c, r)] = p. \tag{26}$$

Since $\pi_\theta(c|q, y)$ is optimized independly of reasoning policy $\pi_\theta(y|q)$, the expectation loss $\mathbb{E}[c|q]$ converges without affect $\pi_\theta(y|q)$ explicitly.

Thus,

$$\mathbb{E}[c|q] = \mathbb{E}_{y \sim \pi_\theta(\cdot|q)}[R(y)]. \tag{27}$$

The detailed proof is provided in Appendix A.3. □

Theorem 5.1 establishes that decoupled confidence estimation yields statistically consistent uncertainty estimates without interfering with policy optimization.

## 6. Experiments

We conduct extensive experiments on mathematical reasoning and code generation tasks, demonstrating that our proposed method improves confidence calibration while preserving reasoning accuracy.

### 6.1. Experimental Settings

**Datasets.** For math reasoning tasks, all RL methods are trained on DeepScalar dataset and evaluated on a suite of mathematical reasoning benchmarks, including MATH-500, AIME 2024/2025 and AMC 2023/2024, which enable joint assessment of reasoning accuracy and confidence calibration.

For code generation tasks, all methods are trained on the PrimeIntellect-verifier dataset (Brown, 2025) and evaluated on widely used benchmarks, including LiveCodeBench v5/v6 and HumanEval+.

**Baselines.** We compare DCPO with representative methods spanning standard RL, post-hoc calibration, and calibration-aware RL: 1) **GRPO** (Shao et al., 2024): standard GRPO objective, serving as the primary baseline for reasoning performance; 2) **ConfClass**: a post-hoc MLP confidence

*Table 2.* Accuracy and calibration evaluation over 3 code generation benchmarks.

| Method | LCB-v5 | | | | LCB-v6 | | | | HE+ | | | |
|---|---|---|---|---|---|---|---|---|---|---|---|---|
| | Acc ↑ | ECE ↓ | PCE ↓ | AUROC ↑ | Acc ↑ | ECE ↓ | PCE ↓ | AUROC ↑ | Acc ↑ | ECE ↓ | PCE ↓ | AUROC ↑ |
| Base | 0.228 | 0.482 | 0.478 | 0.634 | 0.196 | 0.458 | 0.397 | 0.659 | 0.902 | 0.077 | 0.068 | 0.802 |
| GRPO | **0.320** | 0.393 | 0.522 | 0.693 | 0.278 | 0.386 | 0.411 | 0.788 | 0.944 | 0.083 | 0.083 | 0.821 |
| ConfClass | 0.320 | 0.462 | 0.401 | 0.685 | 0.278 | 0.397 | 0.373 | 0.772 | 0.944 | 0.102 | 0.113 | 0.839 |
| RLCR | 0.267 | 0.231 | 0.227 | 0.785 | 0.248 | 0.290 | 0.285 | 0.829 | 0.931 | 0.036 | 0.035 | 0.907 |
| DCPO | 0.314 | **0.223** | **0.205** | **0.796** | **0.283** | **0.267** | **0.250** | **0.845** | **0.947** | **0.034** | **0.032** | **0.914** |

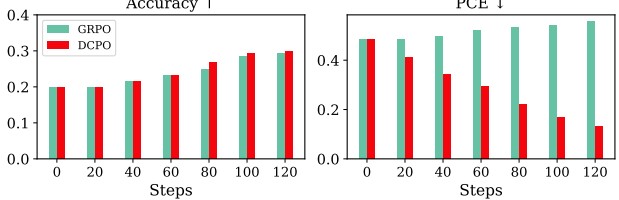

*Figure 6.* Accuracy and PCE on AIME25 dataset at different training steps for GRPO and DCPO. The figures illustrate that during the training process, our method can significantly reduce over-confidence while preserving accuracy.

predictor trained on token-level generation statistics; 3) **RLCR** (Damani et al., 2025): which incorporates a Brier Score calibration loss into the reward; 4) **CCGSPG** (Liu et al., 2025): which modifies the GRPO objective according to token-based confidence. All methods are implemented on widely used LLM Qwen3-8B (non-thinking) to ensure fair comparison.

**Evaluation Metrics.** We evaluate models from 3 complementary perspectives: we use accuracy for reasoning performance, ECE and AUROC for calibration evaluation, and PCE to assess over-confidence..

**Hyperparameters** For math tasks, all models are trained with a global batch size of 256, group rollout size 8, for 3 epochs (approximately 120 steps). For code generation tasks, all models are trained with a global batch size of 128, group rollout size 8, for 3 epoches (approximately 90 steps). Evaluation is conducted with temperature 0.7, top-$p$ 0.8, and top-$k$ 20, following the official Qwen3-8B evaluation protocol (Yang et al., 2025). Detailed training and evaluation parameters are in Appendix B.

## 6.2. Overall Performance

Table 1 and Table 2 reports reasoning accuracy together with calibration metrics on math reasoning tasks and code geenration tasks. Together, these results demonstrate that decoupling correctness and confidence optimization is critical to achieve well-calibrated confidence without degrading reasoning ability across both domains.

**Coupled optimization methods suffer from an accuracy-calibration tradeoff.** As shown in Table 1 and Table 2,

RLCR reduces PCE on AIME24 from 0.510 (GRPO) to 0.214, but at the cost of accuracy drop from 40.0% to 32.8%. Similar trend is observed for CCGSPG. On code generation benchmarks, RLCR reduces ECE on LiveCodeBench v5 from 0.393 to 0.231, but causes accuracy drop from 0.320 to 0.267. These results indicates that directly incorporating calibration losses into the RL objective interferes with correctness-driven policy optimization.

**Post-hoc calibration is insufficient for correcting mis-calibration.** As shown in Table 1, adding classifier on top of GRPO yields only marginal ECE improvements (e.g. from 0.370 to 0.363 on AIME24), while achieving a low AUROC of 0.642, substantially below DCPO's 0.914, which indicates that RLVR significantly distorts the model's internal representation, making post-hoc methods ineffective at substantially reducing mis-calibration.

**DCPO achieves a favorable accuracy-calibration trade-off and reduces over-confidence.** DCPO jointly improves calibration and preserves reasoning performance across both mathematical reasoning and code generation tasks. For example, on AIME24, DCPO achieves 41.6% accuracy, which is comparable to GRPO while reducing PCE from 0.505 to 0.212. On code generation benchmarks, DCPO attains the highest average accuracy (0.515), slightly outperforming GRPO (0.514), while substantially reducing average ECE from 0.287 to 0.175. Figure 6 further illustrates the training dynamics: GRPO exhibits increasing over-confidence during RL, with PCE rising from 0.483 to 0.556 on AIME25, whereas DCPO consistently suppresses over-confidence while maintaining stable accuracy. These results demonstrate that DCPO effectively mitigates RL-induced over-confidence and achieves a substantially better accuracy-calibration tradeoff.

## 6.3. Ablation Studies

We conduct ablation studies to quantify the contribution of each key component in DCPO. Table 3 reports the averaged accuracy and calibration performance across five mathematical benchmarks.

**Effectiveness of decoupled optimization.** Removing decoupled optimization causes degradation across all variants.

*Table 3.* Ablation results averaged over 5 mathematical benchmarks.

| Variant | Acc ↑ | ECE ↓ | PCE ↓ |
|---|---|---|---|
| DCPO | **60.8** | **0.128** | 0.126 |
| *w/o* Instance-level Labels | 60.5 | 0.209 | 0.229 |
| *w/o* Group-level Labels | 58.7 | 0.138 | **0.122** |
| *w/o* Decoupled Optimzation | 57.3 | 0.258 | 0.247 |
| *w/o* On-policy Training | 56.3 | 0.223 | 0.210 |

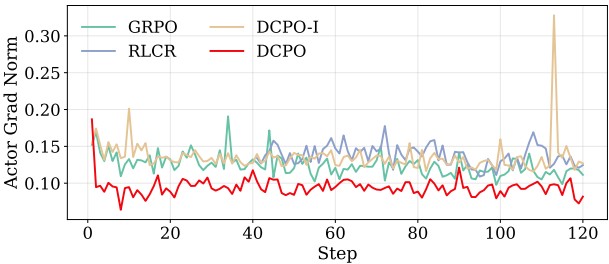

*Figure 7.* The gradient-norm dynamics across different training methods, which demonstrates that DCPO achieves more stable optimization dynamics than other methods.

In particular, ECE increases from 0.128 to 0.258, while accuracy drops from 60.8% to 57.3%, which indicates strong gradient interference in coupled optimization. In contrast, DCPO results in a substantially better accuracy-calibration tradeoff. These results confirm that decoupling is essential for stabilizing training and preventing calibration objectives from harming reasoning performance.

**Effectiveness of hybrid group-instance supervision.** As shown in Table 3, removing instance-level labels leads to a notable increase in ECE from 0.128 to 0.209, while removing group-level labels causes a clear accuracy drop from 60.8% to 58.7%, indicating that group-level accuracy provides a low-variance supervision signal that stabilizes optimization, whereas instance-level correctness enables finer-grained confidence differentiation. By jointly leveraging both signals, DCPO achieves the lowest ECE (0.128) while maintaining the highest accuracy (60.8%), demonstrating the effectiveness of hybrid calibration supervision.

**Importance of on-policy calibration.** As shown in Table 3, off-policy calibration reduces accuracy from 60.8% to 56.3% and increases ECE to 0.223, which demonstrates that off-policy training can interfere with previously learned reasoning behaviors. In contrast, DCPO preserves reasoning performance while achieving better calibration.

Overall, these results demonstrate that *decoupled optimization*, *hybrid group-instance supervision*, and *on-policy calibration* are all critical components of DCPO, and jointly contribute to its superior accuracy-calibration tradeoff.

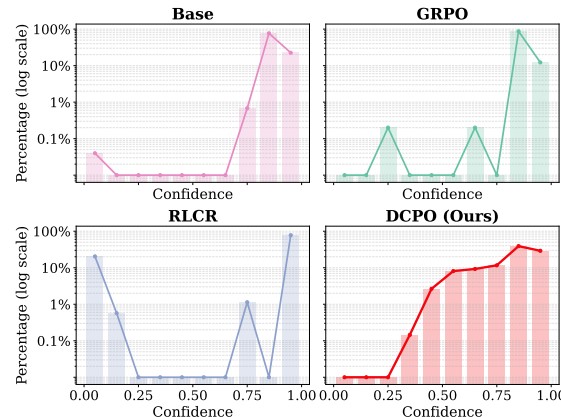

*Figure 8.* Distribution of verbalized confidence predictions across 5 mathematical benchmarks. The y-axis is log-scaled to better visualize the highly concentrated confidence distributions.

### 6.4. Detailed Analysis

**DCPO enables stable optimization dynamics.** To analyze how different calibration strategies affect optimization behavior, we track the $\ell_2$ norm of policy gradients in training. Figure 7 compares the gradient norm trajectories of GRPO, RLCR, DCPO-I, and DCPO. In particular, instance-level calibration methods, such as RLCR and DCPO-I, exhibit pronounced gradient norm fluctuations across training steps, indicating high-variance and unstable optimization. In contrast, DCPO maintains a smoother and more stable gradient norm profile throughout training, which demonstrates that DCPO achieves more stable optimization dynamics.

**DCPO introduces balanced confidence distribution.** To analyze how different training strategies shape verbalized confidence behavior, we visualize the predicted confidence distribution in Figure 8. We observe that GRPO-trained model exhibit heavily skewed confidence distributions, reflecting severe over-confidence. RLCR collapses confidence estimates toward extreme values, indicating poor confidence granularity. In comparison, DCPO produces a balanced and continuous confidence distribution, which demonstrates that decoupled calibration with hybrid supervision is critical for learning expressive and reliable confidence.

## 7. Conclusion

In this paper, we analyze in theory that RLVR inherently induces over-confidence. Moreover, we identified a gradient conflict between accuracy and calibration optimization. Based on these insights, we propose DCPO, a decoupled confidence-aware policy optimization framework. Experiments demonstrate that DCPO significantly improves calibration while preserving reasoning performance. Our study highlights the importance of decoupled optimization in model calibration.

## Acknowledgements

We sincerely thank the reviewers for their insightful comments and valuable suggestions. This work was supported by the Natural Science Foundation of China (No. 62536008, 62476265, 62306303). The authors would like to thank Huawei Ascend Cloud Ecological Development Project for the support of Ascend 910 processors.

## Impact Statement

This paper presents work whose goal is to advance the field of Machine Learning. There are many potential societal consequences of our work, none of which we feel must be specifically highlighted here.

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

# A. Detailed Proofs for Theoretical Analysis

## A.1. Proof of Proposition 1 (Mode Collapse under Trajectory-Level RL)

**Proposition A.1.** *Let $\mathcal{Y}$ denote the finite or countable set of all possible response trajectories for a fixed input $x$, and let*

$$\Delta(\mathcal{Y}) := \left\{ p : \mathcal{Y} \to [0,1] | \sum_{y \in \mathcal{Y}} p(y) = 1 \right\}$$

*denote the probability simplex over trajectories. let $\mathcal{Y}^+ = \{y \in \mathcal{Y} | R(y) = 1\}$ denote the correct trajectories.*
*In the absence of regularizon terms, the optimal solution to RL math problem objective*

$$\max_{p \in \Delta(\mathcal{Y})} \sum_{y \in \mathcal{Y}^+} p(y)$$

*assigns probability mass 1 to a single trajectory $y^* \in \mathcal{Y}^+$.*

*Proof.* Consider the constrained optimization problem

$$\max_{p(y) \geq 0} \sum_{y \in \mathcal{Y}^+} p(y) \quad \text{s.t.} \quad \sum_{y \in \mathcal{Y}} p(y) = 1.$$

Introduce the Lagrangian

$$\mathcal{L}(p, \lambda, \{\mu_y\}) = \sum_{y \in \mathcal{Y}^+} p(y) + \lambda \left( 1 - \sum_{y \in \mathcal{Y}} p(y) \right) + \sum_{y \in \mathcal{Y}} \mu_y p(y),$$

where $\lambda \in \mathbb{R}$ is the multiplier associated with the equality constraint and $\mu_y \geq 0$ are the multipliers associated with the non-negativity constraints.

The Karush-Kuhn-Tucker (KKT) conditions require that, at an optimal solution $p^*$, the following hold:

$$\frac{\partial \mathcal{L}}{\partial p(y)} = \mathbb{I}[y \in \mathcal{Y}^+] - \lambda + \mu_y = 0, \quad \forall y \in \mathcal{Y}, \tag{28}$$

$$\mu_y \geq 0, \quad p^*(y) \geq 0, \tag{29}$$

$$\mu_y \, p^*(y) = 0. \tag{30}$$

We first show that $p^*$ must satisfy $\text{supp}(p^*) \subseteq \mathcal{Y}^+$. Suppose by contradiction that there exists $y \notin \mathcal{Y}^+$ with $p^*(y) > 0$. Then by complementary slackness, $\mu_y = 0$, and from (28) we obtain $-\lambda = 0$, implying $\lambda = 0$. However, for any $y' \in \mathcal{Y}^+$, (28) then yields $1 - \lambda + \mu_{y'} = 1 + \mu_{y'} > 0$, which contradicts stationarity. Hence, $p^*(y) = 0$ for all $y \notin \mathcal{Y}^+$.

Next, consider $y \in \mathcal{Y}^+$. If $p^*(y) > 0$, then by complementary slackness $\mu_y = 0$, and (28) implies $\lambda = 1$. Therefore, for all $y \in \mathcal{Y}^+$, the KKT conditions are satisfied whenever $p^*(y) > 0$.

This shows that any probability distribution supported entirely on $\mathcal{Y}^+$ satisfies the KKT conditions and hence achieves the same optimal objective value. However, such solutions form a face of the simplex $\Delta(\mathcal{Y})$ whose extreme points are precisely the Dirac measures $\delta_y$ for $y \in \mathcal{Y}^+$. Since any distribution with support size greater than one can be written as a non-trivial convex combination of other feasible points, it is not an extreme point of the feasible set.

Consequently, any optimal solution attained at an extreme point assigns probability mass 1 to a single trajectory $y^* \in \mathcal{Y}^+$. $\qquad\square$

This proof result holds irrespective of the size or structure of $\mathcal{Y}^+$, and can easily extend to expective-reward objectives with bounded linearity rewards.

We further note that such low-entropy solutions are stable under small perturbations of the input. Since the policy logits are continuous functions of $x$, extreme logit margins at training inputs induce neighborhoods in the input space where

$$\max_y \pi_\theta(y \mid x') \approx 1, \tag{31}$$

As correctness is not a continuous function of the input, this yields over-confident but incorrect predictions under distribution shift.

### A.2. Proof of Proposition 2 (Gradient Conflict between Correctness and Calibration)

**Proposition A.2** (Natural-Gradient Conflict under Over-Confidence)**.** *Fix an input $x$ and a policy $\pi_\theta(y \mid x)$ defined on a finite or countable action space $\mathcal{Y}$. Define*

$$J_{\mathrm{acc}}(\theta) := \mathbb{E}_{y \sim \pi_\theta(\cdot \mid x)}[R(y)], \qquad Conf_\theta(x) := \mathbb{E}_{y \sim \pi_\theta(\cdot \mid x)}[\phi(y)],$$

*where $R(y), \phi(y) \in [0, 1]$.*

*Let*

$$J_{\mathrm{cal}}(\theta) := -l(Conf_\theta(x), \, J_{\mathrm{acc}}(\theta)),$$

*where $l(c, t)$ is continuously differentiable, convex in $c$, and satisfies*

$$\frac{\partial l}{\partial c}(c, t) > 0 \quad \text{whenever } c > t.$$

*Let*

$$g(y) := \nabla_\theta \log \pi_\theta(y \mid x), \qquad F := \mathbb{E}_{y \sim \pi_\theta}[g(y)g(y)^\top]$$

*denote the score function and the Fisher information matrix.*

*Assume that:*

1. *(Over-confidence)*
$$Conf_\theta(x) > J_{\mathrm{acc}}(\theta).$$

2. *(Positive reward-confidence correlation)*
$$\mathrm{Cov}_{\pi_\theta}(R(y), \phi(y)) > 0.$$

*Then, letting*

$$\langle a, b \rangle_{F^{-1}} := a^\top F^{-1} b$$

*denote the Fisher-metric inner product, we have*

$$\left\langle \nabla_\theta J_{\mathrm{acc}}(\theta), \, \nabla_\theta J_{\mathrm{cal}}(\theta) \right\rangle_{F^{-1}} < 0,$$

*whenever $\nabla_\theta Conf_\theta(x) \neq 0$.*

*Proof.* By the policy gradient theorem,

$$\nabla_\theta J_{\mathrm{acc}}(\theta) = \mathbb{E}_{y \sim \pi_\theta}[R(y)g(y)], \qquad \nabla_\theta Conf_\theta(x) = \mathbb{E}_{y \sim \pi_\theta}[\phi(y)g(y)].$$

Since $\mathbb{E}_{y \sim \pi_\theta}[g(y)] = 0$, these can be written as

$$\nabla_\theta J_{\mathrm{acc}}(\theta) = \mathbb{E}[(R - \mathbb{E}R)g], \qquad \nabla_\theta Conf_\theta(x) = \mathbb{E}[(\phi - \mathbb{E}\phi)g].$$

By Assumption 3 and the chain rule,

$$\nabla_\theta J_{\mathrm{cal}}(\theta) = -\frac{\partial l}{\partial c} \nabla_\theta Conf_\theta(x).$$

Under Assumption 1, we have $\frac{\partial l}{\partial c} > 0$.

Define the natural gradient direction of the confidence as

$$u := F^{-1} \nabla_\theta Conf_\theta(x).$$

Then

$$\langle \nabla_\theta J_{\text{acc}}(\theta), \nabla_\theta Conf_\theta(x) \rangle_{F^{-1}} = \nabla_\theta J_{\text{acc}}(\theta)^\top F^{-1} \nabla_\theta Conf_\theta(x).$$

Substituting the gradient expressions yields

$$\mathbb{E}[(R - \mathbb{E}R)g]^\top F^{-1} \mathbb{E}[(\phi - \mathbb{E}\phi)g] = \text{Cov}_{\pi_\theta}(R(y), \phi(y)),$$

which is strictly positive by Assumption 2.

Therefore,

$$\langle \nabla_\theta J_{\text{acc}}(\theta), \nabla_\theta J_{\text{cal}}(\theta) \rangle_{F^{-1}} = -\frac{\partial l}{\partial c} \, \text{Cov}_{\pi_\theta}(R(y), \phi(y)) < 0,$$

which completes the proof. □

## A.3. Proof of Theorem 1 (Optimality of Decoupled Confidence Estimation)

**Proposition A.3.** *Let $c \in [0, 1]$ denote the predicted confidence, and let $l(c, r)$ be a strictly proper scoring rule. Then the minimizer of*

$$\mathbb{E}_{y \sim \pi_\theta(\cdot|x)} \mathbb{E}_{c \sim \pi_\theta(\cdot|x,y)}[l(c, R(y))]$$

*satisfies*

$$\mathbb{E}[c|x] = \mathbb{E}_{y \sim \pi_\theta(\cdot|x)}[R(y)]$$

*Proof.* For fixed $x$, with policy $\pi_\theta(y|x)$, the correctness reward $R(y)$ is Bernouli distribution with mean

$$p^* = \mathbb{E}_{y \sim \pi_\theta}[R(y)].$$

For a proper calibration estimation function $l(c, r)$ which reaches its minimum at $c = r$, the expected loss

$$\mathbb{E}_{R \sim Bern(p^*)}[l(c, R)]$$

is uniquely minimized at $c = p^*$.

Since $\pi_\theta(c|x, y)$ is optimized independently of the generation policy $\pi_\theta(y|x)$, the expected confidence $\mathbb{E}[c|x]$ converges to the unique minimizer $p^*$ without affect $\pi_\theta(y|x)$ explicitly.

Therefore,

$$\mathbb{E}[c|x] = \mathbb{E}_{y \sim \pi_\theta}[R(y)].$$

□

Decoupled confidence estimation yields statistically consistent calibration without altering the optimality conditions of the generation policy.

## A.4. Proof of Proposition 3 (Unbiasedness and Variance of Group-Level Accuracy)

**Proposition A.4.** *Let*

$$\{y_i\}_{i=1}^G \sim \pi_\theta(\cdot|x)$$

*be indenpendent sampling in a GRPO group, and define*

$$\tilde{R}_G = \frac{1}{G} \sum_{i=1}^G R(y_i).$$

*Then $\tilde{R}_G$ is an unbiased estimator of $\mathbb{E}[R(y)]$ with variance $O(1/G)$.*

*Proof.* Unbiasedness straightforward follows:

$$\mathbb{E}[\tilde{R}_G] = \frac{1}{G} \sum_{i=1}^{G} \mathbb{E}[R(y_i)] = \mathbb{E}[R(y)].$$

Since $R(yi)$ are i.i.d. Bernouli variables,

$$Var(\tilde{R}_G) = \frac{1}{G^2} \sum_{i=1}^{G} Var(R(y_i)) = \frac{1}{G} Var(R(y)).$$

Thus, the variance decays at rate $O(1/G)$. □

**Proposition A.5.** *When using the absolute calibration loss $\ell(c, r) = |c-r|$, using group-level correctness $\tilde{R}_G$ as supervision reduces gradient variance compared to instance-level correctness.*

*Proof.* Let $R(y) \in \{0, 1\}$ denote the instance-level correctness indicator. We assume $R(y)$ is an i.i.d. Bernoulli random variable with
$$\mathbb{E}[R(y)] = p := \tilde{R}_G,$$
where $\tilde{R}_G$ denotes the group-level accuracy.

**Subgradient of the absolute loss.** The absolute loss is not differentiable at $c = r$. Let $g(c, r) \in \partial_c |c - r|$ be an arbitrary measurable selection from the subdifferential

$$\partial_c |c - r| = \begin{cases} \{1\}, & c > r, \\ [-1, 1], & c = r, \\ \{-1\}, & c < r. \end{cases}$$

For any valid selection, we have $|g(c, r)| \le 1$. In the following, we restrict to $c \in (0, 1)$ and $c \ne p$, so that $c \ne r$ almost surely.

**Gradient variance with instance-level supervision.** When supervising with instance-level correctness $R(y)$, the subgradient $g(c, R(y))$ is a random variable taking values in $\{-1, 1\}$. Since $R(y) \sim \text{Bernoulli}(p)$, we have

$$\mathbb{P}(g(c, R(y)) = 1) = 1 - p, \qquad \mathbb{P}(g(c, R(y)) = -1) = p.$$

Therefore,

$$\mathbb{E}[g(c, R(y))] = 1 - 2p, \qquad \mathbb{E}[g(c, R(y))^2] = 1,$$

and the variance is

$$\text{Var}(g(c, R(y))) = 1 - (1 - 2p)^2 = 4p(1 - p).$$

**Gradient variance with group-level supervision.** When supervising with group-level correctness $\tilde{R}_G = p$, the loss reduces to $\ell(c, \tilde{R}_G) = |c - p|$. For $c \ne p$, the corresponding subgradient $g(c, \tilde{R}_G)$ is deterministic, which implies

$$\text{Var}(g(c, \tilde{R}_G)) = 0.$$

**Comparison.** Combining the above results, we obtain

$$\text{Var}(g(c, R(y))) = 4p(1 - p) \ge 0 = \text{Var}(g(c, \tilde{R}_G)),$$

with equality if and only if $p \in \{0, 1\}$, i.e., when all instances in the group are either correct or incorrect.

This completes the proof. □

| Parameter | Value |
|---|---|
| Base model | Qwen3-8B |
| Max prompt length | 1024 |
| Max response length | 3000 |
| Total epochs | 5 |
| Learning rate | $1 \times 10^{-6}$ |
| Training batch size | 256 |
| KL loss coefficient | 0 |
| Rollout group size ($G$) | 8 |
| Rollout temperature | 1.0 |
| Clip ratio (low) | 0.20 |
| Clip ratio (high) | 0.28 |

*Table 4.* Key hyperparameters used in reinforcement learning training.

## B. Training and Evaluation Details

### B.1. Training Setup

All reinforcement learning experiments are conducted using the **VERL** framework on a cluster of **8 NVIDIA A100 (80GB)** GPUs. Table 4 summarizes the key hyperparameters used throughout our experiments.

For **GRPO** and **CCGSPG**, models are trained directly on the original DeepScaler dataset. For methods requiring explicit confidence supervision (e.g., **RLCR** and **DCPO**), prompts are modified to instruct the model to output a confidence score following its final answer. The tradeoff hyperparameter $\lambda$ in DCPO is set to 0.5.

Post-training calibration methods are initialized from the GRPO checkpoint at step 120 and further trained for an additional 80 steps using only Brier Score calibration rewards.

### B.2. Confidence Estimation Baseline

For the **ConfClass** baseline, confidence is predicted using a three-layer MLP with hidden dimension 2048. Training data are generated by sampling outputs from the GRPO-120 checkpoint on the DeepScaler dataset over 20 independent runs. Each sample includes the top-20 token log-probabilities as input features, with the corresponding group-level average accuracy as supervision. The model is trained using cross-entropy loss with a learning rate of $1 \times 10^{-4}$.

### B.3. Evaluation Protocol

Unless otherwise specified, all results are obtained from the 120-step checkpoint of each training run.

Confidence is extracted in two ways:

- **(logits)**: computed as the sequence generation probability by aggregating token-level log-probabilities during decoding.

- **(verbal)**: extracted from model outputs explicitly prompted to provide a confidence score following the `<conf>` tag.

During evaluation, responses are sampled using temperature 0.7, top-$p$ 0.8, and top-$k$ 20 (Yang et al., 2025). To reduce variance on smaller evaluation sets (e.g., AIME24, AMC23), each evaluation is repeated four times with independent sampling, and results are averaged.

Task accuracy is computed using the **OpenCompass** framework. Calibration metrics, including Expected Calibration Error (ECE) and Brier Score (BS), are calculated directly from confidence–accuracy pairs. AUROC is computed using the standard **scikit-learn** implementation.

*Table 5.* Mathematical reasoning performance across different model families and sizes.

| Model | Method | MATH | | AIME24 | | AIME25 | | AMC23 | | AMC24 | |
|---|---|---|---|---|---|---|---|---|---|---|---|
| | | Acc | ECE | Acc | ECE | Acc | ECE | Acc | ECE | Acc | ECE |
| Llama3.1-8B-Instruct | Base | 0.214 | 0.481 | 0.033 | 0.710 | 0.000 | 0.645 | 0.089 | 0.635 | 0.078 | 0.670 |
| Llama3.1-8B-Instruct | GRPO | 0.363 | 0.351 | 0.050 | 0.731 | **0.044** | 0.582 | 0.133 | 0.593 | 0.115 | 0.492 |
| Llama3.1-8B-Instruct | RLCR | 0.288 | 0.214 | 0.033 | 0.372 | 0.000 | 0.387 | 0.100 | 0.301 | 0.107 | 0.398 |
| Llama3.1-8B-Instruct | DCPO | **0.365** | **0.203** | **0.050** | **0.367** | 0.033 | **0.335** | **0.133** | **0.297** | **0.122** | **0.371** |
| Qwen3-14B | Base | 0.893 | 0.068 | 0.325 | 0.476 | 0.250 | 0.452 | 0.657 | 0.189 | 0.578 | 0.267 |
| Qwen3-14B | GRPO | **0.950** | 0.051 | 0.416 | 0.368 | 0.383 | 0.310 | 0.767 | 0.113 | 0.700 | 0.241 |
| Qwen3-14B | RLCR | 0.926 | 0.036 | 0.383 | 0.224 | 0.333 | **0.228** | 0.717 | **0.084** | 0.633 | 0.128 |
| Qwen3-14B | DCPO | 0.942 | **0.033** | **0.433** | **0.211** | **0.383** | 0.230 | **0.775** | 0.088 | **0.692** | **0.114** |

## B.4. Verbalized Confidence Prompts

To enable explicit confidence prediction, training and evaluation prompts are augmented with instructions requiring a floating-point confidence output between 0 and 1. The prompt format is as follows:

> Please put your final answer within `\boxed{}`.
> Also output your confidence for this answer after `<conf>` as a floating-point number between 0 (complete uncertainty) and 1 (full confidence).
> **Example**: `<conf>` `Confidence:` `0.83`
> Ensure that the confidence output appears on a single line and strictly follows the specified format.

## C. Additional Experiments

### C.1. Generalization Across Model Families and Scales

To further evaluate the generalization ability of DCPO, we conduct additional experiments on multiple model families and scales, including Llama3.1-8B-Instruct and Qwen3-14B (non-thinking). Training and evaluation settings follow those in the main paper, and all models are trained on the DeepScaler dataset.

As shown in Table 5, across different model families and scales, DCPO consistently improves calibration while maintaining competitive or improved accuracy. These results further demonstrate the robustness and generalization ability of DCPO across architectures and parameter scales.

### C.2. Sensitivity to Structured Prompts

DCPO relies on structured confidence outputs during training. To ensure that this requirement does not introduce instability, we incorporate a structural penalty of $-1.0$, which is substantially larger than the calibration reward magnitude and thus strongly discourages malformed outputs.

*Table 6.* Format violation rate during training steps.

| Step | 0 | 20 | 40 | 60 | 80 | 100 | 120 |
|---|---|---|---|---|---|---|---|
| Violation Rate | 1.60% | 0.58% | 0.20% | 0.21% | 0.17% | 0.20% | 0.15% |

To further evaluate robustness, we measure the frequency of format violations throughout training. As shown in Table 6, the violation rate is already low at initialization (1.60%) and rapidly decreases to approximately 0.2% after only a few training steps. This trend indicates that the model quickly learns the required output structure and maintains stable formatting behavior during training.

Overall, these results suggest that DCPO is robust to structured prompt requirements, and the structural penalty effectively enforces formatting consistency without introducing instability.

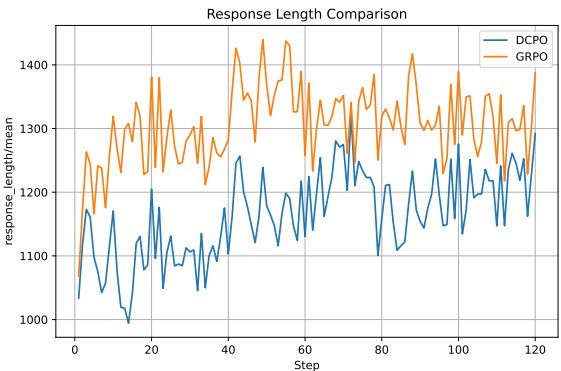

*Figure 9.* Generation length during training.

## C.3. Hyperparameter $\lambda$ Sensitivity

DCPO introduces a hybird coefficient $\lambda$ between group-level and instance-level calibration objectives. In the main paper, we report comparisons among DCPO ($\lambda = 0.5$), DCPO-I ($\lambda = 0$), and DCPO-G ($\lambda = 1.0$), showing that $\lambda = 0.5$ achieves a favorable balance between accuracy and calibration.

*Table 7.* Sensitivity to $\lambda$ in DCPO.

| $\lambda$ | Acc | ECE |
|---|---|---|
| 0 | 0.587 | 0.138 |
| 0.25 | 0.594 | 0.130 |
| 0.5 | 0.608 | **0.128** |
| 0.75 | **0.611** | 0.167 |
| 1 | 0.605 | 0.209 |

To further investigate sensitivity, we conduct additional experiments by varying $\lambda$ across a wider range. The results are summarized in Table 7. We observe that $\lambda$ values between 0.25 and 0.5 achieve the best calibration, while larger $\lambda$ improve training stability by reducing gradient variances. These results suggest that $\lambda = 0.5$ provides the best tradeoff. Larger values of $\lambda$ improve training stability but lead to degraded calibration performance.

These results indicate that DCPO is relatively robust to the choice of $\lambda$ within a reasonable range, and $\lambda = 0.5$ provides a strong tradeoff between accuracy, calibration, and training stability.

## C.4. Impact on Output Length

Since DCPO introduces verbalized confidence outputs, one potential concern is that this additional supervision may increase generation length and computational overhead. To analyze this effect, we measure the average output length of models trained with GRPO and DCPO.

*Table 8.* Average generation length.

| Method | Base | GRPO | DCPO |
|---|---|---|---|
| Length | 1033.63 | 1299.87 | 1292.21 |

Table 8 reports the average generation length, while Figure 9 illustrates the training dynamics. We observe that both GRPO and DCPO increase generation length compared to the base model, which is expected due to reinforcement learning optimization. However, DCPO produces responses with similar or slightly shorter length compared to GRPO.

These results suggest that incorporating verbalized confidence introduces negligible overhead, and DCPO maintains comparable generation efficiency to standard RL training.

