# OpenReview forum: "Decoupling Reasoning and Confidence: Resurrecting Calibration in Reinforcement Learning from Verifiable Rewards"
_ICML.cc/2026/Conference — ICML 2026 regular_

### Official Review · Reviewer_Piy2 · 2026-03-08

**Soundness:** 3
**Presentation:** 3
**Significance:** 3
**Originality:** 3
**Overall Recommendation:** 4
**Confidence:** 1

**Summary:**

This paper was motivated by the fact that maximizing accuracy and optimizing over calibration at the same time might induce instability, as these two are usually negatively correlated. To resolve this issue, this paper proposes to decouple the reasoning and the calibration. They propose the algorithm DCPO, which could decouple these two processes.

They verify the algorithm DCPO in some math reasoning datasets, and the algorithm demonstrates competitive performance compared to existing algorithms.

**Compliance With Llm Reviewing Policy:**

Affirmed.

**Final Justification:**

The rebuttal addressed some of my main concerns. However, based on the novelty, the writing of this paper and the technical contribution of this paper. I will maintain my score of weak accept.

**Key Questions For Authors:**

I have identified the following questions regarding this paper:

1. Could you please provide more insights or intuition (ideally from a theoretical perspective) on how this kind of decoupling works?
2. How can the decoupling resolve the issue of gradient conflict?

**Limitations:**

yes

**Strengths And Weaknesses:**

Strength:

1. This paper is well written.
2. The introduction of the problem setting is clear, and the results and assumptions are stated clearly.
3. The numerical experiments are promising and demonstrate that the algorithms work well in practice.

Weakness:

1. The theoretical results are too simple. They did not show any specific theoretical analysis of why the proposed DCPO algorithm performs well.

---

> ### Author Rebuttal · Authors · 2026-03-31
>
> We sincerely thank the reviewer for the positive assessment and constructive feedback. We are encouraged that the reviewer finds the paper well-written and technically solid. Below, we provide additional theoretical intuition for the decoupling strategy and clarify how DCPO resolves gradient conflicts.
> > **W1: The theoretical results are too simple.**
>
> Due to space limitations, we provide only a brief proof in Section 4, while the complete and rigorous mathematical derivation is presented in Appendix A (spanning more than four pages).
> To further address your concern, we herein provide a detailed explanation of why DCPO performs well:
> - **Why previous methods suffer from gradient conflicts:** In Section 4.2, we show that the gradient direction for maximizing accuracy is negatively aligned with the gradient direction for minimizing calibration error. In particular, under the common over-confidence regime, Proposition 4.2 establishes:
> $ \langle \nabla J_{\mathrm{acc}}, \nabla J_{\mathrm{cal}} \rangle_{F^{-1}} < 0$,
> This result implies that the two objectives are antagonistic under the natural gradient geometry. Consequently, joint optimization leads to destructive gradient interference, causing accuracy–calibration tradeoff. This provides a theoretical explanation for why previous coupled approaches struggle to simultaneously improve both objectives.
> - **Why DCPO mitigates gradient conflicts:** DCPO addresses this issue through a structural decoupling of optimization variables. Specifically, reasoning tokens and confidence tokens are treated as distinct subspaces of policy parameterization. We assign separate reward signals to each component and apply block-wise masked gradient updates. Formally, $\nabla_{\theta}J = \nabla_{\theta}J_r + \nabla_{\theta}J_c,$ where each term is restricted to disjoint token subsets. From an optimization perspective, DCPO effectively converts a conflicting multi-objective problem into two weakly coupled subproblems, thereby improving convergence behavior.
>
> The above theoretical analyses jointly demonstrate why DCPO can achieve the best calibration performance and substantially mitigate the over-confidence issue while preserving accuracy on par with GRPO, and our experimental results in Section 5 further support the validity of these theoretical findings.
>
> > **Q1: Could you provide more insights or intuition on how decoupling works?**
>
> Our decoupling strategy is motivated by the gradient conflict identified in Section 4.2, where accuracy and calibration objectives become negatively aligned when the model is over-confident.
> Intuitively, this conflict arises because accuracy optimization pushes probability mass toward high-reward trajectories (increasing confidence), while calibration penalizes over-confidence and encourages uncertainty awareness. Joint optimization therefore forces the model to both increase and decrease confidence simultaneously, leading to unstable updates.
>
> DCPO resolves this by structurally separating and routing gradients. We partition the output into a reasoning block and a confidence block, and apply block-wise masked updates so that accuracy gradients only affect reasoning tokens, while calibration gradients only affect confidence tokens.
>
> This effectively converts a single conflicting optimization problem into two non-interfering subproblems, allowing reasoning accuracy and calibration quality to improve simultaneously without interference.
>
> > **Q2: How does decoupling resolve gradient conflict?**
>
> Our analysis in Section 4.2 shows that gradient conflict arises from joint optimization, where accuracy and calibration gradients are negatively aligned, leading to the accuracy–calibration tradeoff observed in prior work.
>
> **DCPO resolves this by isolating gradient updates via masked optimization.** Instead of applying both objectives to the full sequence, accuracy gradients are restricted to reasoning tokens, while calibration gradients are restricted to confidence tokens.
> This removes destructive interference by ensuring that the two objectives operate on disjoint parameter pathways.
> Furthermore, Theorem 5.1 guarantees that the confidence predictor remains statistically consistent under strictly proper scoring rules, meaning that decoupled calibration can converge to well-calibrated estimates without affecting reasoning optimization.
>
> Together, these results show that decoupling eliminates gradient conflict while preserving both accuracy and calibration performance.

---

> > ### Author Rebuttal · Reviewer_Piy2 · 2026-04-03
> >
> > Thanks for the response. However, due to the limit of the theoretical results in this paper, I will maintain my original weak accept score.

---

> > > ### Author Response · Authors · 2026-04-07
> > >
> > > We thank the reviewer again for the thoughtful feedback and for maintaining a positive overall assessment.
> > >
> > > Regarding the concern about theoretical depth, we would like to clarify the intended role of our theoretical analysis. Our goal is not to fully characterize the optimization dynamics or provide complete performance guarantees, which we agree would require substantially more involved analysis. Instead, our theoretical results serve two focused purposes:
> > >
> > > 1. to formally identify the existence of gradient conflict between accuracy and calibration objectives (Sec. 4.2), and
> > > 2. to justify why the proposed decoupling strategy can eliminate this conflict through structural separation of optimization pathways.
> > >
> > > We believe this level of analysis is sufficient to explain the design principles of DCPO while keeping the framework general and practically applicable. **Importantly, the reviewer also noted that the method is technically solid and empirically effective. We therefore hope the current theoretical results can be viewed as a principled and sufficient foundation rather than a limitation.**
> > >
> > > We appreciate the reviewer’s feedback and will further clarify the scope and intent of our theoretical analysis in the final version.

---

### Official Review · Reviewer_P19s · 2026-03-10

**Soundness:** 3
**Presentation:** 3
**Significance:** 2
**Originality:** 2
**Overall Recommendation:** 4
**Confidence:** 3

**Summary:**

This paper studies the calibration degeneration problem in RLVR (Reinforcement Learning from Verifiable Rewards), where models tend to become overly confident after reinforcement learning. The authors argue that jointly optimizing accuracy and calibration leads to a gradient conflict. Based on this observation, the paper proposes Decoupled Calibration Policy Optimization (DCPO), which separates reasoning tokens and confidence tokens and applies token-level masked gradients to decouple their optimization objectives. Experiments on several mathematical reasoning benchmarks show that DCPO reduces calibration error while maintaining accuracy comparable to GRPO. Overall, the paper focuses on the overconfidence issue in RLVR and proposes a simple training strategy to mitigate it.

**Compliance With Llm Reviewing Policy:**

Affirmed.

**Final Justification:**

The responses have resolved my concerns. I have raised my score to a positive score.

**Key Questions For Authors:**

1. The method does not appear to improve reasoning accuracy itself. Should the contribution be framed more precisely as improving calibration without harming reasoning performance?
2. Since the model is explicitly trained to predict a verbalized confidence token and calibration metrics are computed based on this token, how can we be sure that the improvements reflect genuine uncertainty estimation rather than simply better prediction of the confidence target?
3. Would the proposed approach remain effective in RLHF settings where rewards are noisier or less structured?

**Limitations:**

yes

**Strengths And Weaknesses:**

Strengths:
1. The problem is meaningful. Overconfidence after RLVR training has been widely observed in recent work. Investigating calibration degeneration is therefore relevant for building more reliable language models.
2. The method is simple and practical. DCPO introduces token-level gradient masking to decouple reasoning and confidence optimization signals. The method does not require architectural changes or additional data.
3. Empirical results support improved calibration. Across several mathematical reasoning benchmarks, DCPO achieves lower calibration error while maintaining similar accuracy compared to GRPO.

Weaknesses:
1. Limited novelty. The core idea of DCPO is to prevent calibration loss from interfering with reasoning token optimization via token-level masking. Conceptually, this appears more like a training strategy refinement for calibration-aware RL rather than a fundamentally new learning framework.
2. Confidence objective may collapse to predicting average accuracy. From the loss formulation, the confidence objective appears close to predicting the average correctness within a rollout group. In such cases, the model may reduce calibration loss simply by outputting values close to the global accuracy, without learning meaningful instance-level uncertainty.
3. No improvement in reasoning capability. The method does not modify the reward or learning signal for reasoning tokens, and therefore does not directly improve reasoning ability. The contribution is better characterized as improving confidence calibration rather than reasoning.
4. Limited experimental validation. The paper motivates calibration as critical for Trustworthy AI in high-risk domains (e.g., healthcare, law, finance), yet experiments are limited to mathematical reasoning tasks with verifiable rewards. The uncertainty structure in such tasks differs substantially from real-world decision-making scenarios, making it unclear whether the method would generalize to practical applications.
5. Weak theoretical support. The paper claims a gradient conflict between accuracy and calibration. However, the theoretical analysis relies on strong assumptions and mainly provides an intuitive explanation rather than a rigorous demonstration that such conflicts arise in practical RLVR training.

---

> ### Author Rebuttal · Authors · 2026-03-31
>
> We thank the reviewer for the thoughtful feedback and for recognizing the importance of calibration degeneration in RLVR. We address each concern below and will incorporate clarifications and additional analysis in the revision.
> > **W1: Limited novelty**
>
> We clarify that **DCPO is not a heuristic refinement, but a theoretically motivated decoupled optimization framework derived from our analysis.**
> 1. **Theoretical identification of calibration degeneration:**
> Section 4.1 shows RLVR training induces mode collapse and systematic overconfidence. Proposition 4.2 further reveals a gradient conflict between accuracy and calibration:$$⟨\nabla J_{acc},\nabla J_{cal}⟩_{F^{-1}}<0,$$
> This explains the empirical accuracy–calibration trade-off observed in prior RLVR-based methods and is validated experimentally (Sec. 3.3).
> 2. **Principled decoupled optimization:**
> Existing approaches jointly optimize accuracy and calibration, leading to gradient interference. Motivated by our analysis, DCPO introduces block-wise verbalized confidence rollout and token-level masked gradients, which are directly derived from the theoretical findings of over-confidence(Prop 4.1) and gradient conflict(Prop 4.2).
>
> Therefore, DCPO is a principled optimization framework rather than a heuristic training strategy.
> > **W2&Q2: Confidence Collapse**
>
> **Both method design and empirical evidence show that DCPO avoids confidence collapse.**
> 1. Method design: DCPO combines group-level and instance-level supervision via a hybrid reward:
> $R_{IG} = λ\tilde{R_{G}} + (1 - λ)R_{i}$,
> where group-level accuracy provides low-variance supervision (Prop. 4.3) and instance-level accuracy preserves sample discrimination, preventing convergence to global-average confidence.
> 2. Experimental evidence: Table 1 shows instance-level calibration metrics of DCPO. Metrics like ECE and AUROC directly evaluate instance-level confidence quality, showing meaningful uncertainty estimation.
> 3. Distribution analysis: Section 6.4(Figure 8) further supports this. Specifically, RLCR and GRPO collapses to limited confidence values, whereas DCPO yields well-separated, continuous confidence distributions, indicating expressive uncertainty estimation.
>
> Overall, DCPO avoids confidence collapse while improving both calibration and accuracy.
> > **W3&Q1: No Improvement in Reasoning**
>
> We respectfully argue that, **as explicitly stated in the Abstract (L26–27) and Introduction (L82–87), DCPO aims to achieve superior calibration while preserving reasoning accuracy comparable to GRPO**. Our experiments support this: compared to base model, DCPO achieves acc gains similar to GRPO (average +11.8%). Achieving further reasoning improvements beyond GRPO is outside the scope of this work.
>
> Moreover, theoretical analysis(Section 4.2) and experiments(Section 3.3, Table 1) suggest that prior calibration methods introduce gradient conflicts that harm reasoning. As shown in Section 3.3, RLCR and CCGSPG improve calibration but reduce accuracy. These results show that DCPO maintains strong reasoning performance while improving calibration.
> > **W4&Q3: Limited Experiment**
>
> **DCPO is a general RLVR training framework applicable across domains**,as it relies on token-level gradient decoupling and task-agnostic confidence prediction. We focus on math because it is the most widely used RLVR domain with standardized evaluation protocols.
>
> Moreover, we currently extend our evaluation to the code generation domain, which has broad real-world, high-risk applications and distinct reward structure from math. Specifically, we use Qwen3-8B, train on PrimeIntellect-verifier dataset, and evaluate on LiveCodeBench v5/v6, and HumanEval+.
>
> | Method | LCB-v5 (Acc/ECE) | LCB-v6 | HE+ |
> |-|-|-|-|
> | Base | 0.228 / 0.482 | 0.196 / 0.458 | 0.902 / 0.077 |
> | DCPO | 0.314 / 0.223 | 0.283 / 0.267 | 0.947 / 0.034 |
>
> DCPO consistently improves both accuracy and calibration, further demonstrating strong generalization.
> These results will be included in the revision.
> > **W5: Theoretical Support**
>
> - **Regarding assumption:** Our proof in Section 4 relies on two assumptions.
>   The overconfidence assumption is supported by our multi-model analysis in Section 3 and prior RLVR studies. Second, the assumption $Cov_{\pi_\theta}(R(y), \phi(y)) > 0$ states higher confidence correlates with higher correctness, consistent with the concept of meaningful confidence estimation and widely observed empirically.
> - **Regarding rigorous proof:** Due to page limits, concise proofs are provided in the main text, with full derivations in Appendix A.
> - **Regarding practical relevance:** Prior methods (e.g., RLCR, CCGSPG) jointly optimize reasoning and calibration, which our theory shows can induce gradient conflicts. Fig. 5 and Table 1 show their accuracy degradation compared to GRPO, providing empirical support.
>
> Overall, we believe that **our theoretical and empirical analyses jointly show the gradient conflict between accuracy and calibration.**

---

> > ### Author Rebuttal · Reviewer_P19s · 2026-04-02
> >
> > Thank you for taking the time to address my concerns. Your responses provided new insights and resolved the issues I had noted. After reviewing the rebuttal, I will consider raising my score.

---

> > > ### Author Response · Authors · 2026-04-07
> > >
> > > Thank you for your thoughtful follow-up and for reconsidering your evaluation. We are glad that our responses provided helpful insights and addressed your concerns. We appreciate your careful review, and we will incorporate these clarifications into the final version to further improve the paper.

---

### Official Review · Reviewer_Jor7 · 2026-03-10

**Soundness:** 3
**Presentation:** 3
**Significance:** 3
**Originality:** 3
**Overall Recommendation:** 4
**Confidence:** 4

**Summary:**

This paper addresses an important problem in RLVR: current models often become excessively over-confident even when producing incorrect answers. The paper argues that this issue is rooted in a fundamental gradient conflict between optimizing for policy accuracy and minimizing calibration error. Based on this insight, they propose the DCPO framework, which decouples reasoning and calibration. Empirical results show that the proposed method preserves reasoning accuracy while achieving the best calibration performance.

**Compliance With Llm Reviewing Policy:**

Affirmed.

**Final Justification:**

I maintain my positive score.

**Key Questions For Authors:**

Please see the weaknesses above. My main question concerns the gap between the theoretical analysis and the practical algorithm design. I would appreciate a clearer explanation of how the theory justifies the final DCPO formulation in practice.

**Limitations:**

Yes

**Strengths And Weaknesses:**

**Strength**

1. The motivation is clearly presented and well justified. The paper thoroughly discusses and empirically demonstrates the over-confidence phenomenon in current LLMs and also shows that existing calibration optimization methods suffer from an accuracy-calibration tradeoff. This provides a strong motivation for the proposed work.

2. The paper provides theoretical analysis for both the over-confidence phenomenon and the accuracy-calibration gradient conflict, which helps support the overall narrative of the paper.

3. Extensive experiments support the main claim that DCPO can consistently suppress over-confidence while maintaining stable accuracy across different benchmarks and settings.

4. The paper is well written and easy to follow.

**Weaknesses**

1. Although the motivation and theoretical analysis are appealing, I believe there is still a gap between the theory and the proposed algorithm that needs to be clarified. In the theoretical analysis, confidence is defined in Equation (7) as an expectation under the model distribution over complete responses. This is also consistent with the related work discussed in Section 2.2, where sequence probability is treated as a confidence signal. However, in the algorithm, the model is instead prompted to generate an explicit confidence output $o_c$, which is a scalar prediction and not the same object as the theoretical confidence quantity. The paper does not provide a formal argument connecting the confidence analyzed in theory to the confidence prediction used in the practical method. As a result, I am not fully convinced that the theory directly supports the final algorithm.

2. Proposition 4.2 is reasonable as a way to show that, under over-confidence, the ascent directions of the accuracy and calibration objectives are negatively aligned. However, the paper should also provide stronger theoretical justification that the proposed DCPO design actually resolves this gradient conflict, beyond the empirical evidence. For example, does the conflict still persist under the actual DCPO objective? Does it specifically arise between reasoning tokens and confidence tokens? Is splitting the output into reasoning tokens and confidence tokens, together with masked gradient updates, necessary or sufficient to address the conflict? At present, the algorithmic design still feels somewhat heuristic, and a more direct theoretical connection would significantly strengthen the paper.

3. I am also concerned that, even with masked gradients, most model parameters remain shared. Therefore, gradients from reasoning-token updates and confidence-token updates can still flow through the same backbone parameters. In that sense, it is not entirely clear whether the method truly decouples reasoning and calibration in parameter space, or whether it only partially separates the supervision signals at the output level. This point deserves further discussion.

---

> ### Author Rebuttal · Authors · 2026-03-31
>
> We thank the reviewer for the thoughtful and constructive feedback. We are encouraged that the reviewer finds the motivation, theoretical analysis, and empirical evaluation strong. Below we clarify the theoretical connection between our analysis and DCPO, and address concerns regarding gradient decoupling.
>
> > **W1:  Gap between theoretical confidence and verbalized confidence**
>
> We clarify that **our theoretical framework does not impose any restriction on the functional form of confidence.** In our formulation, confidence is defined as a general function $Conf_{\theta}(x)$, which can be interpreted as estimating the correctness likelihood of generated response. The analysis in Section 4.2 (Eq. 7) only relieson mild conditions such as over-confidence behavior, and does not assume that confidencee must be derived from sequence probabilities. Moreover,  the verbalized confidence is a parametric scalar prediction conditioned on the input and reasoning trace. Under standard function approximation assumptions, verbalized confidence can serve as a practical surrogate for theoretical confidence[1-2].
>
> Therefore, sequence-probability-based confidence and verbalized scalar confidence are both valid within our general framework. DCPO adopts verbalized confidence primarily because it enables explicit calibration optimization and naturally introduces dedicated confidence tokens for gradient decoupling.
>
> We will clarify this theoretical connection in the revised version.
>
> [1]Corbiere, Charles, et al. "Confidence estimation via auxiliary models." IEEE Transactions on Pattern Analysis and Machine Intelligence 44.10 (2021): 6043-6055.
>
> [2]Kadavath, Saurav, et al. "Language models (mostly) know what they know." arXiv preprint arXiv:2207.05221 (2022).
> > **W2: Does DCPO resolve the gradient conflict?**
>
> **DCPO is designed to mitigate the conflict shown in Section 4.2 by decoupling optimization at the token level.** Existing calibration methods optimize accuracy and calibration over the same trajectory distribution $\pi_\theta(y|x)$, causing gradients to act on the same tokens. In contrast, DCPO separates reasoning tokens and confidence tokens, and applies masked gradient updates:$$\nabla_{\theta}J = \nabla_{\theta}J_r + \nabla_{\theta}J_c,$$where each term applies to disjoint token subsets. As a result, the accuracy and calibration objectives no longer directly compete on the same outputs, which removes the primary mechanism that produces negative alignment. Therefore, token-level separation is not merely heuristic but directly targets the source of gradient conflict identified in Proposition 4.2.
>
> Regarding necessity and sufficiency, token-level decoupling with masked gradients is sufficient to mitigate direct gradient competition between accuracy and calibration objectives. While alternative decoupling strategies may also reduce conflict, DCPO provides a simple and principled mechanism that directly addresses the gradient conflicts.
>
> Together, these results provide a clearer theoretical connection between Proposition 4.2 and DCPO, demonstrating that the proposed design is not heuristic but grounded in resolving the identified gradient conflict. We will revise the paper to make this theoretical connection more explicit.
> > **W3: Does shared backbone still cause gradient interference?**
>
> We agree that DCPO performs functional rather than strict parameter-level decoupling. However, **such decoupling is sufficient to mitigate interference between accuracy and calibration objectives**.
>
> Although parameters are shared, gradients from reasoning tokens and confidence tokens arise from different supervision signals and apply to disjoint output positions. This structured separation encourages the model to utilize different representational subspaces, a phenomenon commonly observed in multi-task learning settings [1–3].
>
> Moreover, full parameter-level decoupling would require an additional model or classifier for confidence prediction. We evaluated such design(Table 1), and found limited improvement in calibration. In contrast, DCPO achieves a favorable accuracy-calibration tradeoff and reduces over-confidence.
>
> Overall, while DCPO does not enforce strict parameter isolation,its token-level supervision and masked gradient updates provides effective and practical decoupling. We will add a more detailed discussion in the revision.
>
> [1]Yu, Tianhe, et al. "Gradient surgery for multi-task learning." Advances in neural information processing systems 33 (2020): 5824-5836.
>
> [2]Wu, Zeqiu, et al. "Fine-grained human feedback gives better rewards for language model training." Advances in Neural Information Processing Systems 36 (2023): 59008-59033.
>
> [3]Yin, Yueqin, et al. "Segmenting text and learning their rewards for improved rlhf in language model." arXiv preprint arXiv:2501.02790 (2025).
>
> We thank the reviewer for the insightful feedback. We believe these clarifications will further strengthen the theoretical grounding of DCPO.

---

> > ### Author Rebuttal · Reviewer_Jor7 · 2026-04-02
> >
> > I thank the author for the detailed rebuttal and will keep my positive score.

---

> > > ### Author Response · Authors · 2026-04-07
> > >
> > > Thank you for the positive feedback and for acknowledging our rebuttal. We appreciate your careful review and are glad that our clarifications addressed your concerns.

---

### Official Review · Reviewer_Vju2 · 2026-03-11

**Soundness:** 3
**Presentation:** 3
**Significance:** 3
**Originality:** 3
**Overall Recommendation:** 4
**Confidence:** 4

**Summary:**

This paper investigates the issues of overconfidence and calibration degradation that arise in large language models (LLMs) during training with verifiable reward-based reinforcement learning (RLVR), specifically GRPO.

Empirical findings: Existing RL methods that couple accuracy with calibration objectives (e.g., RLCR, CCGSPG) inevitably induce an “Accuracy-Calibration Tradeoff.”

Theoretical Insight: Mathematically demonstrates the root cause of this tradeoff—when models exhibit overconfidence, the gradient direction for maximizing accuracy negatively correlates with that for minimizing calibration error (gradient conflict). Simultaneously proves that GRPO's group-level accuracy serves as a calibration supervision signal with lower variance than instance-level accuracy.

Algorithm Design (DCPO): propose an ingeniously decoupled calibration strategy optimization framework. By having the model explicitly output “inference + bounder + confidence” and employing Masked Gradient Optimization during backpropagation, physically isolate gradient updates for accuracy and calibration objectives, thereby resolving the gradient conflict.

**Compliance With Llm Reviewing Policy:**

Affirmed.

**Final Justification:**

All my concerns have been addressed, so I will maintain my positive review.

**Key Questions For Authors:**

# Questions
1. Formatting failures: During the DCPO training process, what percentage of generated trajectories failed to follow the [reasoning]  [score] format? How large was the structural penalty applied in these cases?

2. Hyperparameter λ: How sensitive is the calibration performance to the choice of λ? Did you experiment with dynamically adjusting λ during training (e.g., starting with λ=1 for stability and gradually decreasing to λ=0 to strengthen instance-level discrimination)?

3. Scalability: How does DCPO perform on larger models (e.g., 32B parameters)? Do larger models naturally calibrate better, or do they exhibit the same gradient conflict under RLVR?

4. Impact on generation length: Does appending the verbalized confidence requirement change the length or verbosity of the reasoning trajectories compared to vanilla GRPO?

**Limitations:**

# Limitations
1.	Limited model diversity: Experiments are conducted only on Qwen3-8B. Calibration behavior may vary significantly across different model families (e.g., Llama-3, Mistral, Gemma) and larger-scale models (32B+ parameters).
2.	Domain restriction: Evaluation is entirely focused on mathematical reasoning. The generalizability of DCPO to other tasks, such as code generation or logical puzzles, remains untested.
3.	Sensitivity to structural prompts: DCPO relies on the model correctly outputting the <conf> token and a properly formatted float at the end of the reasoning trajectory. The frequency of format failures and the algorithm’s sensitivity to the structural penalty are not reported.
4.	Lack of hyperparameter sensitivity analysis: The trade-off parameter λ is fixed, and no ablation or sensitivity analysis is provided to show how the balance between low-variance group-level supervision and high-granularity instance-level supervision affects final calibration.
5.	Scalability and generalization: It is unclear how DCPO performs on larger models or in non-mathematical domains. Further testing is needed to confirm whether the method scales effectively and maintains the same benefits.
6.	Potential impact on reasoning trajectory length: Adding a verbalized confidence score may alter the length or verbosity of the generated reasoning steps, which could affect downstream evaluation or efficiency.

**Strengths And Weaknesses:**

# Strengths

## 1. Strong Theoretical Motivation

A key concept explored in this paper is the fundamental mathematical reason behind the trade-off between accuracy and calibration in RL. Proof of Proposition 4.2 (Gradient Conflict): Under over-confident conditions, the inner product, measured by the Fisher norm, between the gradient used to improve accuracy and the gradient used to minimize calibration error is negative. This result is highly elegant and provides a compelling theoretical basis for the proposed masked gradient method.


## 2. Concise and Effective Algorithm Design

The proposed DCPO framework is methodologically rigorous and highly practical.
It employs:
    •    Block-wise verbal confidence rollouts
    •    Token-level gradient masking (Equation 18)

This is a clever implementation that translates theoretical insights into an executable algorithm without requiring multiple independent models or complex structural modifications.

Furthermore, leveraging GRPO’s intrinsic group sampling mechanism to construct a low-variance hybrid calibration objective (Proposition 4.4 / Equation 15) represents an efficient utilization of existing computational resources.

## 3. Convincing Experimental Results

Experiments were conducted on mathematics datasets:
    •    AIME 2024
    •    AIME 2025
	•    AMC 2023
    •    AMC 2024
    •    MATH-500

Table 1 clearly shows:
    •    Compared to baseline methods (such as RLCR and CCGSPG), these methods improve ECE/PCE but sacrifice accuracy significantly;
	•    DCPO substantially improves calibration metrics while maintaining accuracy comparable to (or even slightly higher than) vanilla GRPO;
    •    For example, on AIME24, PCE decreases.

Ablation experiments (Table 2) also effectively validate the necessity of each component.

# Weaknesses
## 1. Limited model diversity

The experiment was conducted on a single model:
    •    Qwen3-8B

Although this is a powerful foundational model, calibration behavior may vary significantly across different model families (e.g., LLaMA-3, Mistral, Gemma) and larger-scale models (e.g., 32B parameters).
Demonstrating that DCPO is equally effective on larger models or different architectures would significantly enhance the paper’s persuasiveness.


## 2. Domain Limitations

The evaluation is strictly limited to mathematical reasoning.

While mathematics is currently the primary test case for RLVR, RLVR is increasingly being applied to:
    •    Code generation (e.g., LiveCodeBench, HumanEval)

Evaluating DCPO on these programming benchmarks would provide stronger evidence of its generalization capabilities.

## 3. Sensitivity to Structured Prompts

DCPO relies on the model correctly outputting the <conf> token and generating a well-formatted floating-point confidence value at the end of the inference trace.

Although the authors mention penalizing structural mismatches, the paper does not report:
    •    the frequency with which the model violates this format during training;
    •    the algorithm’s sensitivity to the magnitude of this penalty coefficient.

This information is crucial for understanding the method’s stability.

## 4. Lack of Hyperparameter Sensitivity Analysis

In Appendix B.1, the balancing parameter used to construct the calibration objective (which balances group-level and instance-level accuracy) is fixed at 0.5.

Providing:
    •    ablation experiments
	•    or sensitivity curves

it would help elucidate how the balance between low-variance supervision and high-granularity supervision affects the final ECE.


## 5. Missing References Related to Confidence Estimation

This section appears to lack relevant literature citations.
We recommend that the authors consider citing:
    •    Confidence is All You Need: Few-Shot RL Fine-Tuning of Language Models

This work investigates the problem of sequence-level confidence estimation.

---

> ### Author Rebuttal · Authors · 2026-03-31
>
> We thank the reviewer for the constructive feedback. We are encouraged that the reviewer finds our theoretical analysis, algorithm design, and empirical results compelling. Below we address each concern and provide clarifications.
> > **W1&Q3: Model Diversity**
>
> We appreciate the suggestion to validate DCPO across more model families and scales. To address the concern, **we conduct additional experiments on Llama3.1-8B-Instruct and Qwen3-14B (non-thinking).**  Training and evaluation hyperparameters follow the main paper, and the training data is based on DeepScaler.
> | Model | Method | MATH-500 (Acc/ECE/AUROC) | AIME24 | AIME25 | AMC23 | AMC24 |
> |-|-|-|-|-|-|-|
> | Llama3.1-8B-Instruct | Base | 0.21 / 0.48 / 0.73 | 0.03 / 0.71 / 0.31 | 0.00 / 0.65 / N/A | 0.09 / 0.64 / 0.37 | 0.08 / 0.67 / 0.34 |
> | Llama3.1-8B-Instruct | DCPO | 0.37 / 0.20 / 0.90 | 0.05 / 0.37 / 0.73 | 0.03 / 0.34 / 0.68 | 0.13 / 0.30 / 0.64 | 0.12 / 0.37 / 0.70 |
> | Qwen3-14B | Base | 0.89 / 0.07 / 0.72 | 0.33 / 0.48 / 0.82 | 0.25 / 0.45 / 0.77 | 0.66 / 0.19 / 0.77 | 0.58 / 0.27 / 0.79 |
> | Qwen3-14B | DCPO | 0.94 / 0.03 / 0.88 | 0.43 / 0.21 / 0.90 | 0.38 / 0.23 / 0.92 | 0.78 / 0.09 / 0.83 | 0.69 / 0.11 / 0.89 |
>
> **Across different model families and scales, DCPO consistently improves both accuracy and calibration, demonstrating strong generalization**. Due to rebuttal time constraints,  more comprehensive experimental results  will be included in the final version.
> > **W2: Domain Limitation**
>
> **We would like to clarify that DCPO is a general training framework applicable to multiple RLVR domains**, including math, code generation, QA, etc. We focus on math because it is widely adopted in prior RLVR research and benefits from well-established baselines and evaluation protocols.
>
> Moreover, **we extend our evaluation to code generation.** Specifically, we train Qwen3-8B on PrimeIntellect-verifier dataset, and evaluate on LiveCodeBench v5/v6, and HumanEval+.
> | Method | LCB-v5 (Acc/ECE) | LCB-v6 | HE+ |
> |-|-|-|-|
> | Base | 0.228 / 0.482 | 0.196 / 0.458 | 0.902 / 0.077 |
> | DCPO | 0.314 / 0.223 | 0.283 / 0.267 | 0.947 / 0.034 |
>
> **DCPO consistently improves in both accuracy and calibration in code generation, further demonstrating its cross-domain generalization capability.** We will include additional experimental results in the final version.
> > **W3&Q1: Sensitivity to Structured Prompts**
>
> We introduce a structural penalty of **−1.0**, which is substantially larger than the calibration reward magnitude, thereby discouraging malformed outputs.
> To further address the concern, we measured the frequency of format violations during training:
> | Step | 0 | 20 | 40 | 60 | 80 | 100 | 120 |
> |-|-|-|-|-|-|-|-|
> | Violations | 1.60% | 0.58% | 0.20% | 0.21% | 0.17% | 0.20% | 0.15% |
>
> The violation rate is low (<2%) at initialization and rapidly drops to ~0.2%, indicating that **the model learns the required structure early in training and that the algorithm is stable with respect to the penalty coefficient.**
> We will include these analyses and discussions in the revision.
> > **W4&Q2: Hyperparameter Sensitivity**
>
> In our experiments(Table 1), we have reported comparisons among DCPO (λ = 0.5), DCPO-I (λ = 0), and DCPO-G (λ = 1.0), showing that λ = 0.5 achieves a balance between accuracy and calibration.
> To further address the concern, **we conducted additional ablation experiments by varying λ:**
> | λ | Acc | ECE |
> |-|-|-|
> | 0  | 0.587 | 0.138 |
> | 0.25 | 0.594 | 0.130 |
> | 0.5 | 0.608 | 0.128 |
> | 0.75 | 0.611 | 0.167 |
> | 1  | 0.605 | 0.209 |
>
> As shown above, λ values between 0.25 and 0.5 achieve the best calibration, while larger λ improve training stability by reducing gradient variances. These results suggest that λ = 0.5 provides the best trade-off.
> We will include these additional ablation results and discussion in the revised paper.
> > **W5: Missing Related Work**
>
> Thank you for pointing this out. We will add discussion of [1] in the related work, specifically, this work focuses on sequence-level confidence estimation and applies confidence signals in RL training, which is complementary to our gradient-decoupling approach.
>
> [1]Li, Pengyi, et al. "Confidence is all you need: Few-shot rl fine-tuning of language models." arXiv preprint arXiv:2506.06395 (2025).
> > **Q4: Impact on generation length**
>
> To address this concern, we present the average output length of models trained with GRPO and DCPO, along with a plot illustrating the dynamics of output length during training for both methods.
> | Method | Base | GRPO (step 120) | DCPO (step 120) |
> |-|-|-|-|
> | Length | 1033.63 | 1299.87 | 1292.21 |
>
> [Response length plot ](https://anonymous.4open.science/r/Figs-B056/rebuttal.png)
>
> As shown above, **DCPO produces similar or slightly shorter response than GRPO, indicating negligible overhead from verbalize confidences.**
>
> In conclusion, we sincerely thank the reviewer for the valuable comments. We hope these responses address the concerns and strengthen the paper.

---

> > ### Author Rebuttal · Reviewer_Vju2 · 2026-04-02
> >
> > Thanks for the feedback, here 3 points:
> > 1. paper wrote CCGSPG, but table wrote CCGPSG
> > 2. "dentify a fundamental" seems typo
> > 3. additional results without baseline, especially code domain

---

> > > ### Author Response · Authors · 2026-04-07
> > >
> > > Thank you for the careful reading and helpful follow-up comments. We address each concern and provide clarifications below.
> > >
> > > > **1&2. Typos**
> > >
> > > We thank the reviewer for pointing out these issues. CCGSPG is the correct method name, and CCGPSG in the table is a typo. Similarly, “dentify a fundamental” should be “We identify a fundamental.” We will correct these and carefully proofread the final version to avoid similar typos.
> > >
> > > > **3. Additional results without baselines**
> > >
> > > Thank you for this valuable suggestion. We agree that including baselines is important for fair and comprehensive evaluation. Due to time and computational resource constraints during the first-round rebuttal, we prioritized reporting improvements of DCPO over the base model in both accuracy and calibration.
> > >
> > > To further address this concern, we have extended our experiments to include additional baselines including GRPO and RLCR, and report the full results below.
> > > **Across all settings, compared with baselines, DCPO demonstrates a consistent and significant advantage in calibration while maintaining competitive or improved accuracy. This trend holds across different model families (Qwen, Llama), scales (8B, 14B), and domains (code and math), indicating strong generalization.**
> > > - For instance, in the code domain, DCPO achieves an average accuracy of 0.515 on Qwen3-8B across 3 benchmarks, comparable to GRPO (0.514), while significantly reducing ECE from 0.287 to 0.175. In contrast, RLCR incurs a noticeable accuracy drop (0.514 → 0.482).
> > > - In the math domain, DCPO matches GRPO in accuracy on Llama3.1-8B-Instruct (0.141) while substantially improving calibration (ECE: 0.550 → 0.315). On Qwen3-14B, DCPO slightly improves accuracy over GRPO and further reduces ECE (0.217 → 0.135), whereas RLCR again trades accuracy for calibration.
> > >
> > > | model     | method | Avg Acc | Avg ECE | LCB v5 Acc | LCB v5 ECE | LCB v6 Acc | LCB v6 ECE | HE+ Acc | HE+ ECE |
> > > |-----------|--------|-------------|-------------|----------------------|----------------------|----------------------|----------------------|----------------|----------------|
> > > | Qwen3-8B | Base | 0.442 | 0.339 | 0.228 | 0.482 | 0.196 | 0.458 | 0.902  | 0.077 |
> > > | Qwen3-8B  | GRPO   | 0.514| 0.287 | **0.320** | 0.393  | 0.278 | 0.386 | 0.944 | 0.083 |
> > > |  Qwen3-8B | RLCR   | 0.482 | 0.186       | 0.267    | 0.231    | 0.248  | 0.290   | 0.931  | 0.036    |
> > > | Qwen3-8B  | DCPO   | **0.515** | **0.175**   | 0.314  | **0.223**    | **0.283**    | **0.267**    | **0.947**   | **0.034**  |
> > >
> > > | Model | Method | Avg Acc | Avg ECE | MATH Acc | MATH ECE | Aime24 Acc | Aime24 ECE | Aime25 Acc | Aime25 ECE | AMC23 Acc | AMC23 ECE | AMC24 Acc | AMC24 ECE |
> > > |-------|--------|---------|---------|--------------|--------------|------------|------------|------------|------------|-----------|-----------|-----------|-----------|
> > > | Llama3.1-8B-Instruct | Base | 0.083 | 0.628 | 0.214 | 0.481 | 0.033 | 0.710 | 0.000 | 0.645 | 0.089 | 0.635 | 0.078 | 0.670 |
> > > | Llama3.1-8B-Instruct | GRPO | 0.141 | 0.550 | 0.363 | 0.351 | 0.050 | 0.731 | **0.044** | 0.582 | 0.133 | 0.593 | 0.115 | 0.492 |
> > > | Llama3.1-8B-Instruct | RLCR | 0.106 | 0.334 | 0.288 | 0.214 | 0.033 | 0.372 | 0.000 | 0.387 | 0.100 | 0.301 | 0.107 | 0.398 |
> > > | Llama3.1-8B-Instruct | DCPO | **0.141** | **0.315** | **0.365** | **0.203** | **0.050** | **0.367** | 0.033 | **0.335** | **0.133** | **0.297** | **0.122** | **0.371** |
> > > | Qwen3-14B | Base | 0.540 | 0.289 | 0.893 | 0.068 | 0.325 | 0.476 | 0.250 | 0.452 | 0.657 | 0.189 | 0.578 | 0.267 |
> > > | Qwen3-14B | GRPO | 0.643 | 0.217 | **0.950** | 0.051 | 0.416 | 0.368 | 0.383 | 0.310 | 0.767 | 0.113 | **0.700** | 0.241 |
> > > | Qwen3-14B | RLCR | 0.600 | 0.140 | 0.926 | 0.036 | 0.383 | 0.224 | 0.333 | **0.228** | 0.717 | **0.084** | 0.633 | 0.128 |
> > > | Qwen3-14B | DCPO | **0.645** | **0.135** | 0.942 | **0.033** | **0.433** | **0.211** | **0.383** | 0.230 | **0.775** | 0.088 | 0.692 | **0.114** |
> > >
> > > We sincerely appreciate the reviewer’s  constructive feedback and  will include the complete experimental results in the final version.

---

### Decision · Program_Chairs · 2026-04-30

**Decision:**

Accept (regular)

**Comment:**

Reviewers agree on strong theoretical motivation (gradient conflict proof), simple yet effective DCPO algorithm (token-level decoupling), and compelling experiments showing preserved accuracy with superior calibration on math benchmarks. Rebuttals effectively addressed concerns: added cross-model (Llama3.1-8B, Qwen3-14B), cross-domain (code: LiveCodeBench, HumanEval+), ablation (λ sensitivity, format violations <0.2%), and length analyses, with DCPO outperforming baselines like GRPO/RLCR in calibration without accuracy loss. All four reviewers recommend weak accept (score 4), with consistent positive final justifications after rebuttals. Therefore, we recommend acceptance.